# Astronomical calibration of the middle Cambrian in Baltica: global carbon cycle synchronization and climate dynamics

Valentin JAMART [1] ✉, Damien PAS [1,2] ✉, Linda A. HINNOV[3],
Jorge E. SPANGENBERG [4], Thierry ADATTE [1], Arne T. NIELSEN [5],
Niels H. SCHOVSBO [6], Nicolas THIBAULT [5], Michiel ARTS[2] &
Allison C. DALEY [1]

The Alum Shale Formation of Baltica preserves one of the most continuous and fossil-rich records for the Cambrian Period. Thus, the Alum Shale is a key sedimentary archive for refining global chronostratigraphy, reconstructing carbon cycle perturbations, and assessing astronomical forcing of high-latitude systems during an early Palaeozoic greenhouse world. Here we present a high-resolution cyclostratigraphic and multiproxy study of the middle Cambrian succession from the Albjära-1 drill core (southern Sweden), anchored by a high-precision U–Pb age. Integration of our astronomical time scale with carbon isotope data and refined biostratigraphy places the Albjära-1 core as a global reference record. This framework provides numerically constrained ages and durations for the Drumian Carbon Isotope Excursion (DICE), enabling worldwide synchronization of biostratigraphy and carbon cycle events. Coupled elemental geochemistry and time calibration reveal that obliquity- and orbital eccentricity-driven climate oscillations modulated sea-level and dust fluxes, highlighting the sensitivity of Earth's early Paleozoic greenhouse systems to astronomical forcing.

The Cambrian Period (538.8–486.85 Ma) represents a fundamental transition in Earth's history. It is marked by rapid evolutionary diversification (Fig. 1A; e.g., Cambrian Explosion, Cambrian Substrate Revolution, and onset of Ordovician Planktonic Revolution), at least ten major carbon isotope excursions (CIEs), and pronounced sea-level variations[1–4]. Together, these events highlight the Cambrian as a key interval for investigating the interplay between biological innovation, environmental change, and global biogeochemical cycles.

Within this framework, the Miaolingian Epoch (Wuliuan to Guzhangian stages) stands out as a period of repeated ecological turnovers and geochemical perturbations. Major events include: the Redlichiid-Olenellid Extinction and associated ROECE carbon isotope excursion at the Series 2 – Miaolingian boundary (Fig. 1A)[2,5]; the shift from endemic to cosmopolitan faunal communities during the Wuliuan (Fig. 1A)[3,6]; the Drumian Carbon isotope Excursion (DICE) coincident with global sea-level rise (Fig. 1A)[1,7–9]; and the extinction of the Marjumiid biomere and Damesellid trilobites during the onset of the Steptoean Positive Carbon isotope Excursion (SPICE) at the Miaolingian – Furongian boundary (Fig. 1A)[2,10].

Despite their global significance, the pacing of Miaolingian events remains poorly constrained. The challenging correlation of the Miaolingian Epoch is due to a number of local and global factors:

[1]Institute of Earth Sciences (ISTE), University of Lausanne, CH-1015 Lausanne, Switzerland. [2]SediCClim, Geology Department, University of Liège, Liège, Belgium. [3]Atmospheric, Oceanic & Earth Sciences Department, George Mason University, 4400 University Drive Fairfax, Virginia, USA. [4]Institute of Earth Surface Dynamics (IDYST), University of Lausanne, CH-1015 Lausanne, Switzerland. [5]Department of Geosciences and Natural Resource Management, University of Copenhagen, Øster Voldgade 10, DK 1350 Kbh K, Denmark. [6]Geological Survey of Denmark and Greenland (GEUS), Øster Voldgade 10, DK-1350 Copenhagen K, Denmark. ✉e-mail: Valentin.Jamart@unil.ch; Damien.Pas@unil.ch

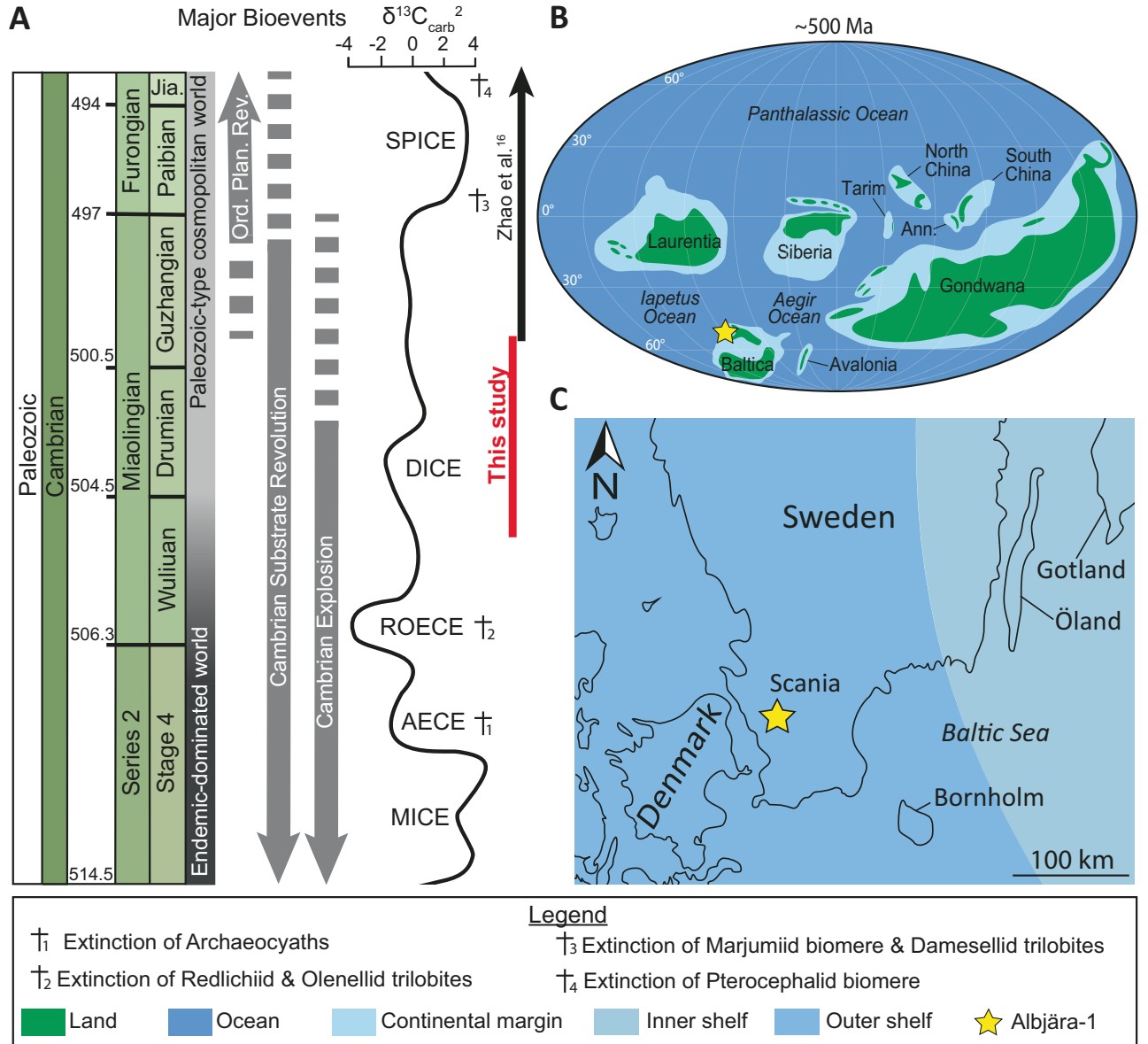

**Fig. 1 | Geological setting. A** Major geochemical event throughout the Series 2 to Furongian showing the studied time interval (red vertical line), modified from Jamart et al.[4]. **B** Global Cambrian paleogeography showing the position of the studied Albjära-1 drill core along the western margin of Baltica modified from Jamart, et al.[4] with adjustment of the position of Avalonia and Gondwana based on Landing et al.[11], Keppie et al.[12], and Park[13]. **C** Geographic location of the Albjära-1 drill site in Scania during the early Drumian (*Ptychagnostus atavus* Biozone), modified from Nielsen and Schovsbo[14]. The stage boundaries are taken from the latest version v2024-12 of the international chronostratigraphic chart updated from Cohen et al.[15]. AECE Archaeocyathid Extinction Carbon isotope Excursion, Ann. Annamia, carb carbonate, CIE Carbon Isotope Excursion, DICE DrumIan Carbon isotope Excursion, Jia. Jiangshanian, Ma Mega annum, MICE MIngxinsi Carbon Isotope Excursion, Ord. Plan. Rev. Ordovician Plankton Revolution, ROECE Redlichiid-Olenellid Extinction Carbon isotope Excursion.

1) disconformities that truncate the geological record; 2) a scarcity of well-preserved outcrops; 3) widespread endemism that characterizes the early and early middle Cambrian; and 4) limited numerical age control for the 497-506.5 Ma interval[1,2,4,5,7].

The Alum Shale Formation (ASF) in Baltica provides a rare opportunity to address these challenges. This organic-rich, outer-shelf black mudstone succession preserves one of the most continuous records of the Miaolingian Series worldwide. The ASF, extending from the Wuliuan *Ptychagnostus gibbus* (agnostoid) to the Tremadocian *Bryograptus kjerulfi* (graptolite) Biozone, has been central to numerous studies in paleontology, stratigraphy, geochemistry, and, more recently, cyclostratigraphy (see Supplementary Information for details). These studies have established a robust biostratigraphic framework for Baltica, but key uncertainties persist at the base of the

Miaolingian Series in Baltica due to (1) truncation of the sedimentary record by the Hawke Bay unconformity *sensu lato*, (2) limited biostratigraphic resolution, and (3) imprecise radioisotope absolute temporal framework for stratigraphic correlation[14,16–18]. In particular, the Hawke Bay unconformity *sensu lato* obscures the precise stratigraphic position and expression of the DICE in Baltica, complicating global chemostratigraphic correlation. Moreover, the mechanisms linking sediment supply to orbital forcing in Baltica remain largely hypothetical[17]. These challenges underscore the need to develop a high-resolution chronometer capable of constraining the Miaolingian succession within the ASF.

Paleoclimatologic research has established that quasi-periodic oscillations in the Sun-Earth position, i.e., the Milankovitch cycles, have induced significant variations in Earth's past climate at different time

scales[19,20]. Cyclostratigraphy is a well-established and powerful chronometer, which uses the expression of Milankovitch cycles preserved in the stratigraphic record to refine the geological time scale[20-24]. The principal frequencies of Milankovitch cycles, in the Cambrian Period, includes short and long orbital eccentricity ( ~ 95-135 and ~405 ka), and their amplitude modulation ( ~ 2.4 Myr); obliquity ( ~ 32 ka), and its modulations ( ~ 173 ka and ~1.2 Myr); and the precession index ( ~ 17 and 20 ka, modulated by orbital eccentricity)[25-30]. Applying cyclostratigraphic methods to the ASF therefore offers a way to construct an astronomically calibrated age model for the Miaolingian succession and to test links between orbital forcing, sediment supply, and carbon-cycle perturbations.

In this study, we apply a high-resolution, multiproxy, cyclostratigraphic approach to the Albjära-1 core from Scania, southern Sweden. Building on the same drill core that refined the chronology of the Furongian Series (147-212.4 m)[16,31,32], we focus on the lower part of the core (212.4-237.4 m) to constrain the Miaolingian dynamics. Our approach integrates (1) a revised biostratigraphy, (2) a 15 cm-resolution $\delta^{13}C_{org}$ dataset, and (3) 1 mm-resolution X-ray fluorescence (XRF) core-scanning data, on which we applied cyclostratigraphic time-series analysis to extract Milankovitch-scale variability across the upper Wuliuan–lower Guzhangian succession.

In this work, we identify and characterize the DICE (one of the ten major carbon isotope perturbations of the Cambrian), refine the location of the Wuliuan–Drumian and Drumian–Guzhangian stage boundaries, and establish an astronomically calibrated time scale (ATS) anchored to the U-Pb age of 499.9 ± 0.9 Ma[16]. The positioning of the DICE along with the numerically time-calibrated astronomical time scale permits synchronizing this event, and thereby the Miaolingian, worldwide. By combining detrital (Ti) and geochemical ratios (Ti/Al and Si/Al) with the ATS, we propose a paleoclimatic model that links sedimentary delivery in Baltica to Milankovitch forcing during the Miaolingian.

## Results and discussion
### Geological setting
The major first-order sea-level rise during the Cambrian led to the development of a vast epicontinental shelf sea that eventually covered nearly the entire Baltic Sea region[16,33,34]. This level-bottomed shelf (Scandinavian Shelf) preserves a thick outer shelf black mudstone succession known as the Alum Shale Formation (ASF)[14,33,34]. In Scania, southern Sweden, the ASF (Fig. 2) is approximately 100 m thick and is characterized predominantly by finely laminated, non-bioturbated, organic-rich black mudstone[16,34,35]. Within the studied interval, the ASF also contains up to 1 m-thick primarily deposited carbonate beds, corresponding to the Andrarum and Exsulans limestones, as well as diagenetic carbonate concretions (anthraconite) that may reach 50 centimeters in thickness.

During the Miaolingian, the Albjära-1 drill site in Scania was located on the outer shelf (below storm wave base) of western Baltica, at ~50–55° southern latitudes (Fig. 1)[33]. The 237.4 m-long Albjära-1 core encompasses the interval from the lower Cambrian to the Upper Ordovician[36]. The Cambrian-earliest Ordovician section of the core represents a time span from the early Cambrian Stage 2 (*Platysolenites antiquissimus* Biozone) to the Tremadocian Stage (*Bryograptus kjerulfi* Biozone), corresponding to the Alum Shale, Gislöv, and Hardeberga formations (see Supplementary Information for details).

### Stratigraphy
The Albjära-1 core is exceptionally well-preserved, with a recovery rate near 100%, and stratigraphically continuous, capturing all middle Cambrian biozones known from Scandinavia[36]. This makes the core a regional reference section of global relevance for the outer shelf of the Miaolingian Series in Baltica.

For clarity and ease of reference, we assign abbreviated labels to two intervals of the core. The interval from 200.0 to 212.2 m,

previously studied by Zhao et al.[16,31,32], is hereafter referred to as ACU (Albjära-1 Core Upper part). The interval from 212.2 to 237.4 m investigated in this study is designated ACL (Albjära-1 Core Lower part).

The combination of literature and updated biostratigraphy enables us to unambiguously identify the Wuliuan – Drumian and Drumian – Guzhangian boundaries in the ACL (Fig. 2 and S2).

The ACL encompasses four key events: 1) the Hawke Bay unconformity *sensu lato* from 232.745 to 232.616 m (93.74-93.611 m AD), 2) a negative excursion, possibly the DICE, from 232.616 to 229.3 m (93.611-90.295 m AD), 3) the Wuliuan – Drumian boundary at 229.3 m (90.295 m AD), defined by the first occurrence of the agnostoid *Ptychagnostus atavus*[1,7-9], and 4) the Drumian – Guzhangian boundary at 215.073 m (76.305 m AD), defined by the first occurrence of the agnostoid *Lejopyge laevigata*[2,38].

In this work, the original drilling depths are referred to as depth (in m), and the Adjusted Depth, corresponding to the depth after reduction of the limestone thickness, is referred to as depth AD (in m), allowing for direct continuity with the adjusted depths presented in Zhao et al.[16].

### Rock-eval pyrolysis
The Rock-Eval results are presented in Supplementary datasets. The TOC content ranges from 0.06 to 8.51 wt.%, with a mean value of 4.35 wt.% (Fig. 2). The S2 peaks of the measured samples range from 0.03 to 0.64 mg HC/g, with a mean value of 0.21 mg HC/g. The HI and OI values, respectively, range from 2 to 97 mg HC/g TOC and 2 to 757 mg CO2/g TOC. The $T_{max}$ values, for S2 peaks ≥ 0.2 mg HC/g, range from 293 to 608 °C with a mean value of 492.69 °C. The carbonate contents of the ASF, excluding limestone layers, range from 0.32 to 10.38 wt.% (Fig. 2).

### Carbon stable isotopes ($\delta^{13}C_{org}$)
The $\delta^{13}C_{org}$ values (‰ vs. VPDB) range from -33.0 to -27.6‰, with an average value of −31.0‰ (Fig. 2; Supplementary datasets). The limestone beds (e.g., Andrarum and Exsulans limestones) display significantly lower values (>1‰) than the general trend observed for $\delta^{13}C_{org}$. These lighter values correspond to significantly high HI values compared to the black mudstone intervals, suggesting a slightly different type and/or preservation of organic matter[39]. Consequently, those primarily limestone-induced outliers (plotted and highlighted in Fig. 2) were removed and are not discussed further.

From 232.616 to 229.975 m (93.611 to 90.97 m AD), the $\delta^{13}C_{org}$ values reach a plateau of mean value of ~−32.8‰. From 229.975 to 224.598 m (90.97 to 85.83 m AD), the $\delta^{13}C_{org}$ values increase from −32.0 to −30.5‰, followed by a 0.5 ‰ decrease in the last 80 cm of the interval. From 224.598 to 213.838 m (85.83–75.07 m AD), the $\delta^{13}C_{org}$ values reach a plateau of mean value of ~−31‰. From 213.838 to 212.22 m (75.07–73.452 m AD), the $\delta^{13}C_{org}$ values shows a saw-tooth pattern, with a slight increase from −31 to −30‰.

The $\delta^{13}C_{org}$ values recorded in the ACL fall within the expected Cambrian range, suggesting a limited diagenetic imprint on the organic carbon isotopes (Fig. S3A)[40,41].

### Preservation of the carbon isotope signal
Baltica has been subject to post-depositional processes, including diagenesis and low-grade metamorphism[42-44], which potentially influenced the preservation of the primary geochemical signal[45,46,48]. The reliability of our geochemical data is essential, and a meticulous evaluation was conducted (see Supplementary Information).

The combined results from the analysis of $T_{max}$, HI/OI ratio, and XRD clay mineral identification confirm that the ACL experienced late diagenesis to very low-grade metamorphic conditions and that the organic matter is mature to overmature, in line with previous observations of Zhao et al.[31], Buchardt and Lewan[42], Sanei et al.[43], and Hammer and Svensen[44].

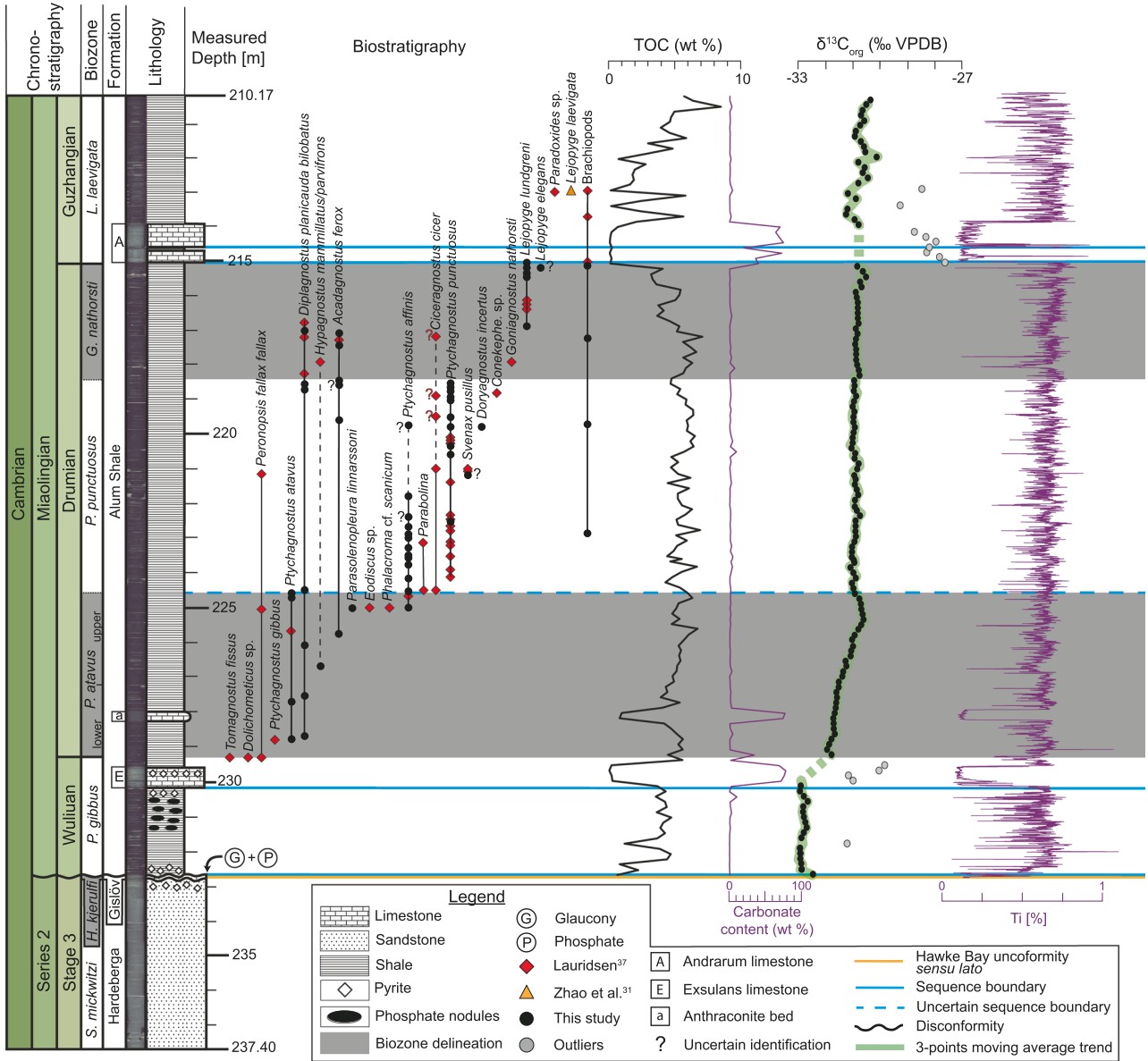

**Fig. 2 | Biogeochemical framework of the Albjära-1 core.** The global chronos-tratigraphic chart is from Cohen et al.[15]. Biostratigraphic data are built on the work of Lauridsen[37] and Zhao et al.[31]. G. *Goniagnostus*, H. *Holmia*, L. *Lejopyge*, org organic, P. *Ptychagnostus*, S. *Schmidtiellus*, TOC Total Organic Carbon, VPDB Vienna Peedee Belmnite, wt weight.

The general trend observed in the isotopic signal remains reliable for chemostratigraphic correlations, even though the absolute value of $\delta^{13}C_{org}$ being slightly changed owing to post-depositional processes[4,47].

### Identification of the DICE event in Baltica

The precise location and identification of the DICE in Baltica is challenging, essentially owing to the Hawke Bay unconformity *sensu lato*. Also, the generally low carbonate content only permits $\delta^{13}C_{org}$ analysis rather than $\delta^{13}C_{carb}$, and there are limited studies on $\delta^{13}C_{org}$ in the Wuliuan – Drumian interval in Baltica. To address this issue and identify the DICE in the ACL, we compared our $\delta^{13}C_{org}$ data with those from Baltica and surrounding areas (see Supplementary Information for more details).

Resulting from these correlations, we propose that the negative $\delta^{13}C_{org}$ excursion in the ACL, characterized by an amplitude of -1.5‰ and a peak value of −33‰, represents the DICE. This excursion extends from the unconformity at the base of the Alum Shale Formation to the

Wuliuan – Drumian boundary, coinciding with the *Ptychagnostus gibbus* Biozone (Fig. 2).

Integration of carbon-isotope stratigraphy with agnostoid biostratigraphy allows clear identification of the DICE in Baltica, predating the lowest occurrence of *Pt. atavus*, thereby refining the global temporal framework for this key Cambrian event.

### Correlations with the global carbon isotope framework

The identification of the five global biozones (*Pt. gibbus, Pt. atavus, Pt. punctuosus, G. nathorsti*, and *L. laevigata*) and the DICE in the ACL enables global correlation of Baltica with sections worldwide and with the global Cambrian biostratigraphic framework (Fig. 3).

A comprehensive review of the literature by Jamart et al.[4] identified 135 localities documenting sedimentary successions spanning the Wuliuan – Drumian interval. Of these, only 23 record the DICE (see Supplementary datasets for details). For global correlation, we selected 10 of these sites, including the DICE and representative of the main

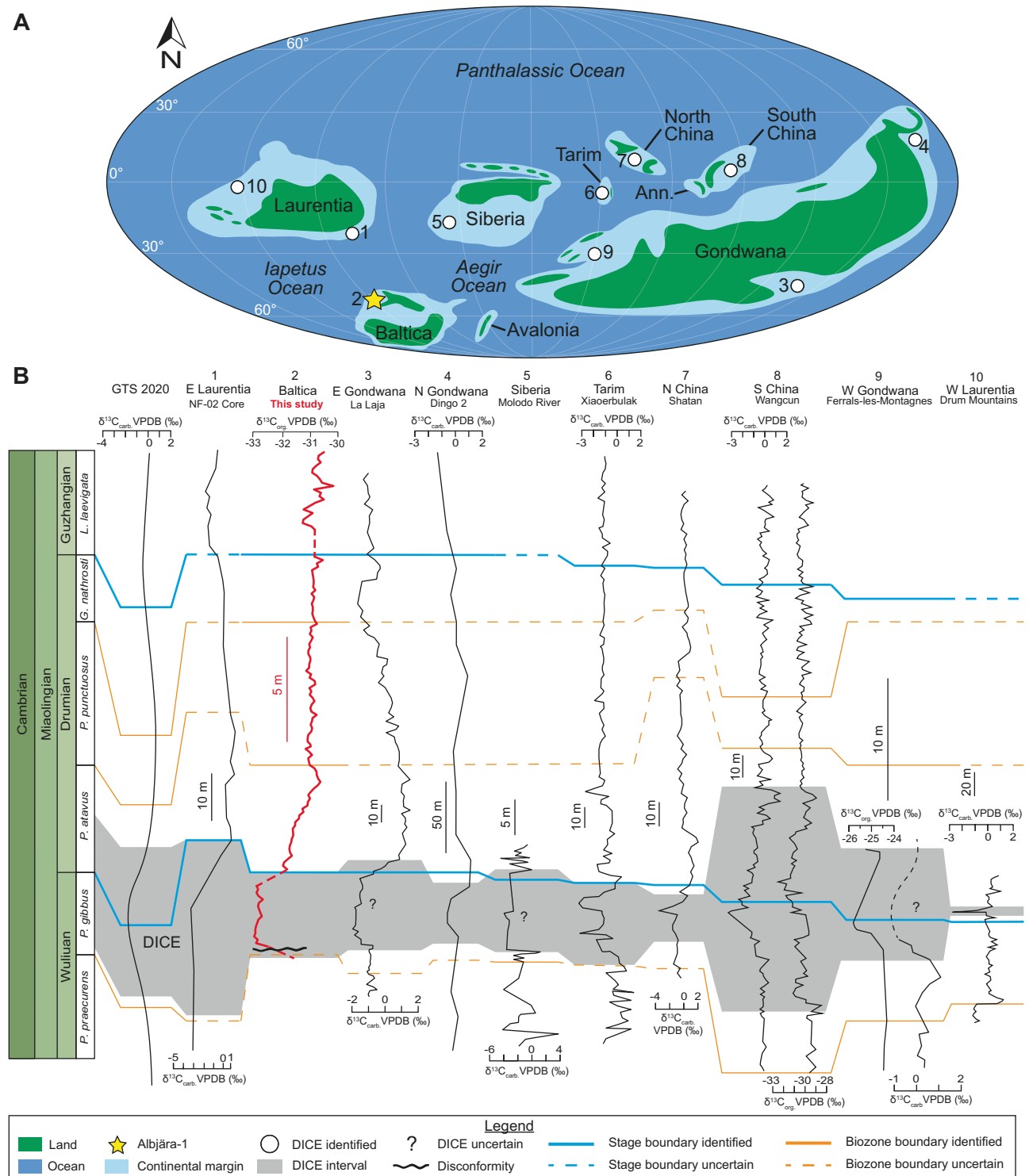

**Fig. 3 | Correlation of the Albjära-1 core (Scania, Baltica) with the global carbon geochemical framework. A** Localities recording the DICE on different paleo-continents, adapted from Jamart et al.[4] with adjustment of the position of Avalonia and Gondwana based on Landing et al.[11], Keppie et al.[12], and Park[13]. **B** δ13C isotope curves from Peng et al.[2], NF-02 core(1)[52], this study (2), La Laja section (3)[53], Dingo 2 well (4)[54], Molodo River section (5)[55], Xiaoerbulak section (6)[56], Shatan section (7)[57], Wangcun section (8)[8], Ferrals-les-Montagnes section (9)[4], and Drum Mountains section (10; Drumian GSSP)[58]. The grey shaded area corresponds to the DICE interval. For a complete list of the localities that recorded the Drumian Stage, see Supplementary datasets. Ann. Annamia, carb carbonate, DICE Drumian Carbon isotope Excursion, G. *Goniagnostus,* GSSP Global Boundary Stratotype Section and Point, GTS Geologic Time Scale, L. *Lejopyge* P. *Ptychagnostus,* org organic, VPDB Vienna Peedee Belmnite.

Cambrian landmasses (Fig. 3; Laurentia, Baltica (this study), Siberia, Gondwana, North and South China, and Tarim).

In the GTS 2020, the DICE peak is correlated with the base of the Drumian Stage, defined by the First Appearance Datum (FAD) of *Pt.*

*atavus.* However, the use of the FAD alone to delineate Cambrian stages and series is debated, as the FADs appear less synchronous than previously assumed[48–50]. In line with chemostratigraphic practice, where peaks of prominent CIEs are correlated between sections[31,51,52],

we aligned the DICE peak among the ten selected sections in which it is recorded (Fig. 3). In seven out of these ten sections, the DICE peak occurs below the lowest occurrence of *Pt. atavus*. This offset may reflect asynchrony in either the onset of the DICE or the first appearance of *Pt. atavus* or both.

## Cyclicity in the Ti series

Well logs recorded in the Albjära borehole (gamma ray (GR) and resistivity; see Supplementary Information) within the ACL interval exhibit patterns consistent with the variations observed in detrital proxies (Ti, Si, Al, K, Zr). This suggests minimal post-depositional alteration (Fig. S5) and supports the use of detrital elements in time-series analysis. For the present study, titanium (Ti) was selected as the primary proxy to facilitate integration with the dataset published by Zhao et al.[16], thereby ensuring continuous coverage across the Miaolingian Series (see Supplementary Information).

In the stratigraphic record ( = depth domain), four main cyclicities are observed in the Ti signal at wavelengths of 0.08–0.1 m, 0.12–0.2 m, 0.42–0.72 m, 0.76–1 m, and possibly 1.75–2.25 m (Fig. 4). The expression of these main cycles varies with depth. In the 90.2-93.6 m interval, the 0.08–0.1 m, 0.12–0.2 m, and 0.76–1 m cycles are present but weak, and the 0.42–0.72 m and 1.75–2.25 m cycles are difficult to recognize. In the 76.4–90.2 m interval, the 0.76–1 m cycle dominates, and the 0.08–0.1 m and 0.12–0.2 m cycles are also well expressed, but the 1.75–2.25 m cycle is uncertain. The 0.42–0.72 m cycle is absent below 87 m but becomes well developed between 76.4-87 m. In the 73.5–76.4 m interval, most of the cycles are weak, although the 0.12–0.2 m cycle is more distinct than the 0.08–0.1 m and 0.42–0.72 m cycles; the 0.76–1 m and 1.75–2.25 m cycles are uncertain.

## Spectral analysis and detection of astronomical metronomes

The $2\pi$ MTM spectrum of the uncalibrated Z-score-normalized Ti series shows 28 frequencies above the 90% confidence level (CL) (Fig. 5A; supplementary datasets). Eleven of these frequencies have ratios consistent with theoretical Cambrian astronomical cycles[16,25,29,30,59], allowing us to define five frequency bands. Four of these bands exceed the 90% CL threshold, while the first band peaks at 80% CL: 1) 0.4 to 0.6 cycles/m (= 1.75–2.25 m), likely corresponding to the 405 kyr long eccentricity ($E_{405}$); 2) 1.03 to 1.32 cycles/m (= 0.76–0.97 m), the 173 kyr obliquity modulation ($I_{173}$); 3) 1.4 to 2.55 cycles/m (= 0.42–0.72 m), the short eccentricity band ($e_{100}$); 4) 5 to 8.5 cycles/m (= 0.12–0.2 m), the obliquity band ($o_{30}$); and 5) 9.8 to 12.5 cycles/m (= 0.08–0.1 m), the precession band ($p_{18}$). The bands match those reported by Zhao et al.[16] for the ACU interval. Amongst the 28 frequencies above 90% CL, 15 fall within the Milankovitch bands and are discussed further. Comparable frequency bands are observed in the spectra of the uncalibrated Z-score-normalized data for the other detrital elements (Fig. S6). Notably, in these elements the 0.4–0.6 cycles/m band reaches 90% CL, although its power remains lower than the 1.03–1.32 cycles/m band.

The 173 kyr and 405 kyr cycles are referred to as the inclination and eccentricity metronomes, respectively[25]. The metronome designation corresponds to astronomical cycles that are considered stable over time and can be used as a chronometer to reliably calibrate sedimentary sequences[25].

Overall, the spectral analysis indicates that the 173 kyr cycle ( = inclination metronome) is the most prominent and persistent signal amongst the detrital elements, whereas the commonly used 405 kyr long-eccentricity cycle ( = eccentricity metronome) is nearly absent in the Ti record. This lack of expression makes the 405 kyr metronome unsuitable for tuning our dataset. The strong power in the obliquity signal is consistent with previous findings of high-latitude astronomical forcing by Zhao et al.[16] and Sørensen et al.[17].

Given the unusually strong expression of the 173 kyr cycle in our dataset, we assessed its reliability and applicability for tuning the Ti

series by applying Average Spectral Misfit (ASM), Multi-taper Method (MTM), Evolutive Harmonic Analysis (EHA), and the frequency ratio methods (Figs. 4, 5, S6–9; see Supplementary Information for more details). The results show that all the methods support the presence of astronomical forcing and indicate that the inclination metronome persists throughout the studied interval.

Assuming that the 173 kyr cycle influenced the ACL depositional record, we used it as a tuning reference to calibrate the Ti series.

The MTM spectrum of the astronomically calibrated Ti series reveals distinct peak power corresponding to $E_{405}$, $I_{173}$ (calibrated), $e_{100}$, $o_{30}$, and $p_{18}$ periodicity bands (Fig. 5B). In the $E_{405}$ band, one frequency with weak power is identified. In the $I_{173}$ band, one frequency with high power is identified as expected when tuning to a single astronomical frequency, minimal tuning *sensu*[60]. In the $e_{100}$ band, five frequencies are identified and correspond to durations of 135.5, 126.7, 120.7, 113.4, and 106.6 kyr. In the $o_{30}$ band, five frequencies are identified and correspond to durations of 39.5, 37.2, 31.9, 31.7, and 31.4 kyr. In the $p_{18}$ band, three frequencies are identified, and these correspond to durations of 20, 16.3, and 15.9 kyr, with the latter two lying below the 90% CL. Overall, the durations observed in the $E_{405}$, $I_{173}$, $e_{100}$, $o_{30}$, and $p_{18}$ bands align with published values[16,25,30], supporting our astronomical interpretation.

To test the presence of the inclination metronome in the Ti series, we examined the phase relationship between the obliquity signal, the 173 kyr cycle, and the -1.2 Myr obliquity modulation (s4-s3) arising from the interaction of Mars and Earth secular frequencies (Fig. 5C; see Supplementary Information for more details). The analysis reveals a strong, consistent phase relationship between the 173 kyr cycle and the -1.2 Myr obliquity modulation throughout the ACL, in line with theoretical predictions[24,25]. This alignment supports the identification of the inclination metronome in the dataset, as these cycles, governed exclusively by planetary motions, are considered relatively stable over the geological timescale[25,61,62].

At shorter timescales, the ACL (1.8–4 Myr) exhibits a phase relationship between obliquity and $I_{173}$, whereas a limited antiphase relationship between the 173 kyr cycle and the 27–40 kyr obliquity band appears in the upper ACL ( - 0–1.8 Myr). Overall, these results indicate that long-term obliquity modulation is preserved, whereas short-term obliquity signals may be partially disrupted (see Supplementary Information for the detailed Hilbert modulation analysis).

The analysis of the tuned data ( = time domain) reveals three intervals in the Ti series (Fig. 6). From 497.085 to -499.4 Ma, the long-term trend is mainly modulated by long and short eccentricity cycles, with obliquity being present but subordinate (Fig. 6A). Between -499.4 to -500.3 Ma, the dominant cycle is difficult to resolve, and this interval is interpreted as a gradual transition from eccentricity- to obliquity-dominated forcing (Fig. 6B). From -500.3 to 504.197 Ma, the obliquity dominates, with the 173 kyr cycle as the principal signal (Fig. 6C).

## Astronomically-tuned age model for the Miaolingian Series

Aligning our age-depth model to the early Guzhangian numerical age of 499.9 ± 0.9 Ma[16] at 73.72 m, we construct a high-resolution astronomical timescale (ATS) for the Miaolingian Series in Baltica. To further reduce the uncertainty of the age model, we tracked the 173 kyr band using Continuous Wavelet Transform (CWT) in four detrital elements (Al, Si, Ti, K).

The ATS provides an estimated duration of 6.920 + 0.386/-0.167 Myr for the Miaolingian Series (Fig. 7). Within this framework, the durations of the Guzhangian (3.165 + 0.259/-0.041 Myr), Drumian (3.009 ± 0.078 Myr), and Wuliuan (part; 0.746 ± 0.049 Myr) stages are resolved. Key biozone and the DICE durations are also quantified: *Pa. forchhammeri* (3.856 + 0.235/-0.065 Myr), *G. nathorsti* (0.691 ± 0.024 Myr), *Pt. punctuosus* (1.326 ± 0.032 Myr), *Pt. atavus* (0.992 ± 0.022 Myr), almost complete *Pt. gibbus* (0.746 ± 0.049 Myr) and *Pa. paradoxissimus* (3.064 ± 0.102 Ma), and the DICE ( - 750-800

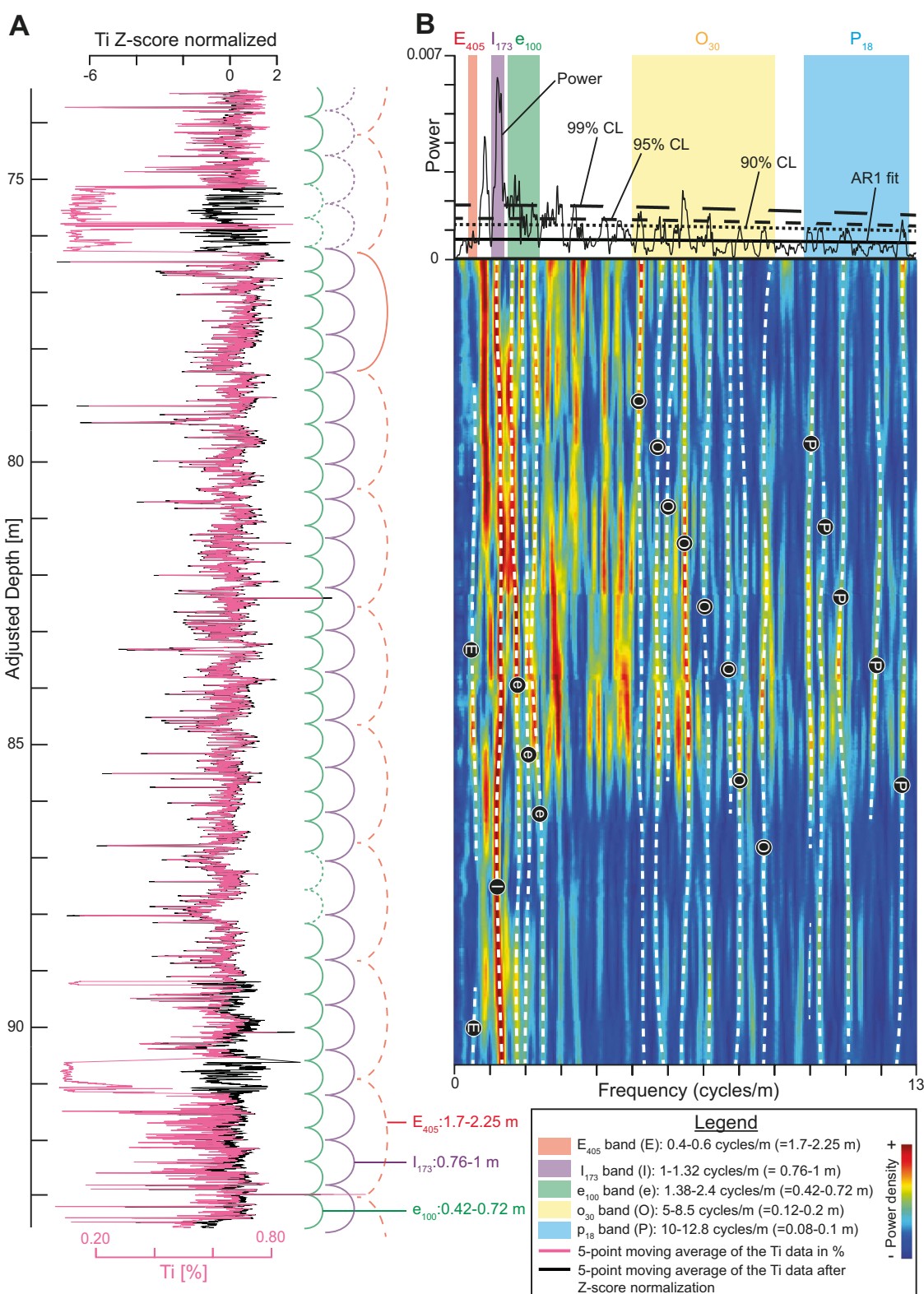

**Fig. 4 | Evaluation of astronomical cycles in the untuned Ti data of the ACL. A** Visual cyclicity observed in the ACL. The green, purple, and red solid lines correspond to the modulating cycles on the long-term trend of the Ti stratigraphic series. The dashed intervals of these lines are meant to highlight absence or weakly expressed cyclicity in the long-term trend of the Ti stratigraphic series. **B** 2π-MTM classic AR-1 and EHA at 5 mm resolution (Ti series Z-score normalized). The data have been detrended using a 20% locally weighted scatterplot smoothing (LOWESS). ACL Albjära-1 Core Lower part, AR1 Autoregressive model, CL Confidence level, E and $E_{405}$ Long Eccentricity, e and $e_{100}$ Short Eccentricity, EHA Evolutive Harmonic Analysis, I and $I_{173}$ 173 kyr cycle, MTM Multi-Taper Method, O and $o_{30}$ Obliquity, P and $p_{18}$ Precession. Further details concerning the 2π-MTM classic AR-1 power spectrum are provided in Fig. 5.

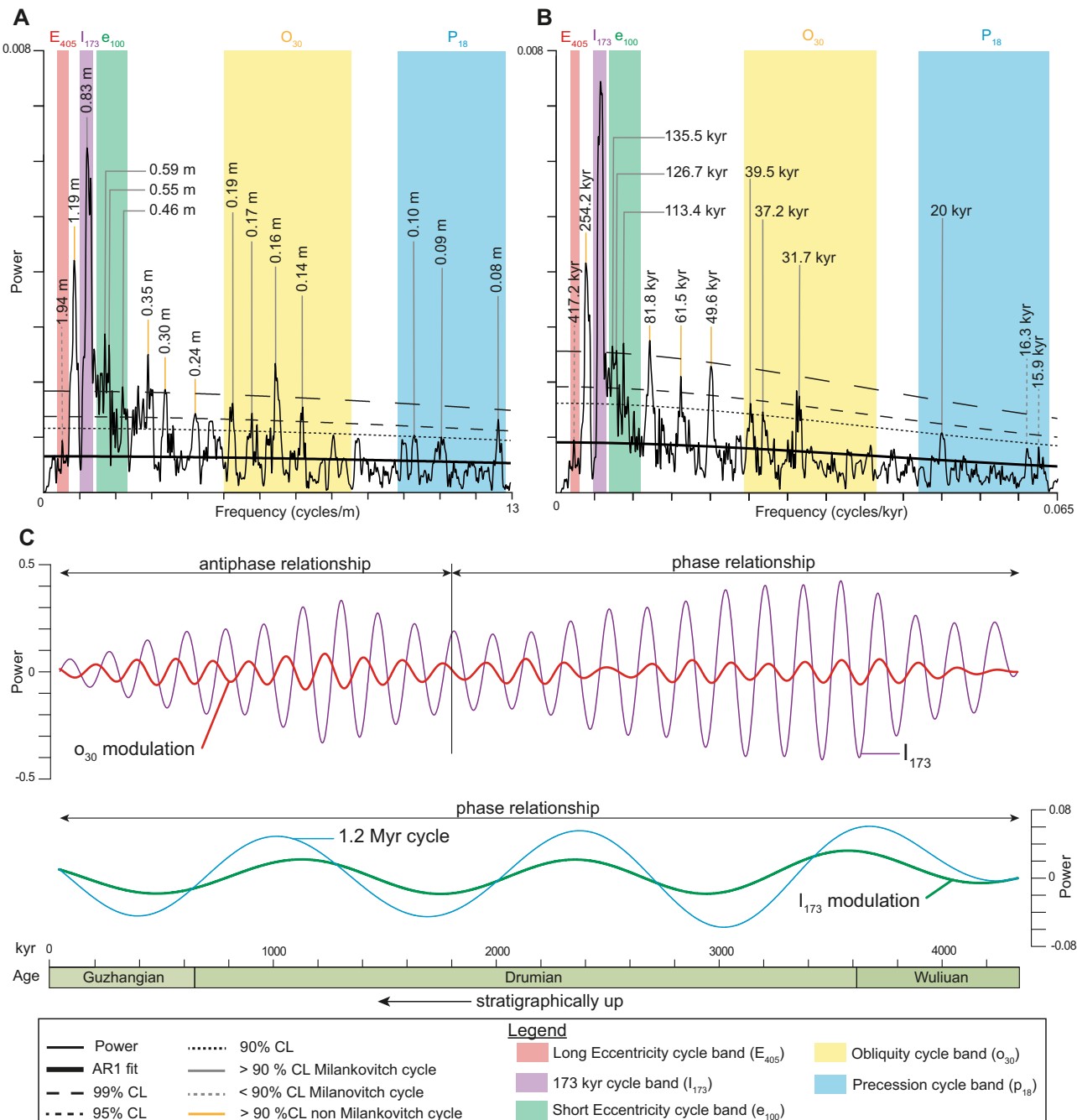

**Fig. 5 | Multi-Taper Method and amplitude modulation analyses.** The dataset is at a 5 mm resolution, and a 20% locally weighted scatterplot smoothing (LOWESS) detrending is applied. **A** 2-π MTM classic AR1 analysis on the untuned dataset in the stratigraphic record (= depth domain). **B** 2-π MTM classic AR1 analysis on the $I_{173}$ metronome-tuned dataset (= time domain). **C** Amplitude modulation analysis of the obliquity band modulation (27–40 kyr) versus 173 kyr cycle (155-195 kyr passband) and 173 kyr modulation (155–195 kyr passband) versus 1.2 Myr cycle (1150-1300 kyr passband) on the non-detrended dataset. AR1 Autoregressive model, CL Confidence Level, $E_{405}$ Long Eccentricity, $e_{100}$ Short Eccentricity, $I_{173}$ 173 kyr cycle, kyr kilo year, MTM Multi-Taper Method, Myr Million year, $o_{30}$ Obliquity $p_{18}$ Precession.

kyr). Our age estimate of 503.45 ± 1.02 Ma and 500.44 ± 0.94 Ma, respectively for the base and the top of the Drumian Stage closely aligns with available U-Pb ages bracketing this stage in West Gondwana (base between 503.14 ± 0.13 Ma in Landing et al.[63] and 505 ± 1 Ma in Palacios et al.[64]; top slightly above 500.9 ± 0.9 Ma in Palacios et al[64] and 500.55 ± 0.9 Ma in Landing et al.[65]) as well as with the U-Pb age of 501.45 ± 0.10 Ma identified in the upper Drumian of Avalonia in Landing et al.[66]. These results provide critical chronostratigraphic constraints for the global Cambrian time scale and enables the Miaolingian Series to be synchronized worldwide. Additional details on biozones,

stages, and events durations, as well as the wavelet analysis, are provided in Supplementary Information.

## Orbital forcing and climate-control mechanisms on detrital input in the Alum Shale Formation

In Scania, climatic fluctuations inferred to have primarily influenced sediment supply into the basin likely involved reworking of inner-shelf sediments, driven by storm activity associated with sea-level changes[14,16–18], and/or enhanced aeolian dust fluxes during obliquity minima[14,16–18,23,34].

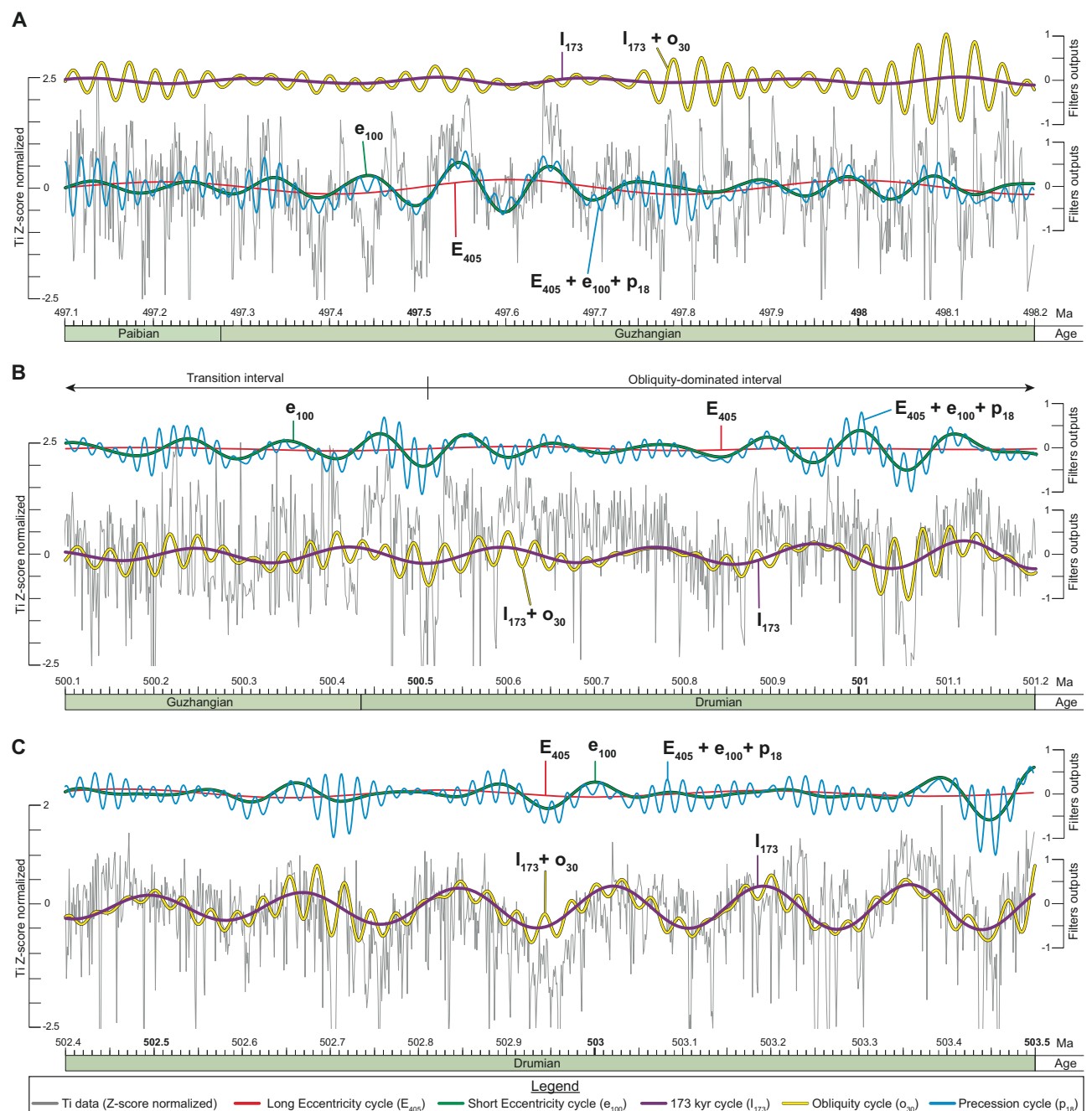

**Fig. 6 | Astronomical forcing in the Albjära-1 core.** Close-up view of selected part of the eccentricity-dominated (**A**), end of the transition zone (**B**), and obliquity-dominated (**C**) intervals observed in the tuned Ti time series (Z-score normalized). To better visualize the influence of the Milankovitch cycles on the Ti signal, the dominant cycles (obliquity-related or eccentricity-related) are shown on the Ti data in intervals (**A**–**C**), while the other cycles are shown above the Ti data. $E_{405}$ Long Eccentricity cycle, $e_{100}$ Short Eccentricity cycle, $I_{173}$ 173 kyr cycle, Ma Mega annum, $o_{30}$ Obliquity cycle, $p_{18}$ P – recession cycle.

Stratigraphic records from both the Cambrian and Ordovician indicate stronger obliquity forcing than observed today[23]. This interpretation is supported by the identification of obliquity-band frequencies and the dominance of a 173 kyr metronome in the ACL signal. Unlike in the Cenozoic, where strong obliquity forcing is linked to glacio-eustasy, the Cambrian records appear to reflect a different mechanism.

The transition from obliquity- to eccentricity-dominated forcing shown in Figs. 6 and 7 may reflect variation in orbital parameters (s3 and s4) that led to the disappearance of the $I_{173}$ in the Ti record, thereby favoring preservation of eccentricity modulation[24], and/or the northward drift of Baltica, which carried the craton farther into the Ferrel-like atmospheric cell, reducing aeolian dust transport to Scania and weakening the obliquity imprint.

## Obliquity-paced aeolian input

In offshore deposits such as the ACL, aeolian dust fluxes from the continent account for a major component of the sedimentary record[67–69]. Vegetation coverage of land was negligible in the early Paleozoic – essentially microbial – resulting in a higher sensitivity of landmasses to physical erosion and transport mechanisms, such as enhanced aeolian sedimentary processes[70]. During the middle

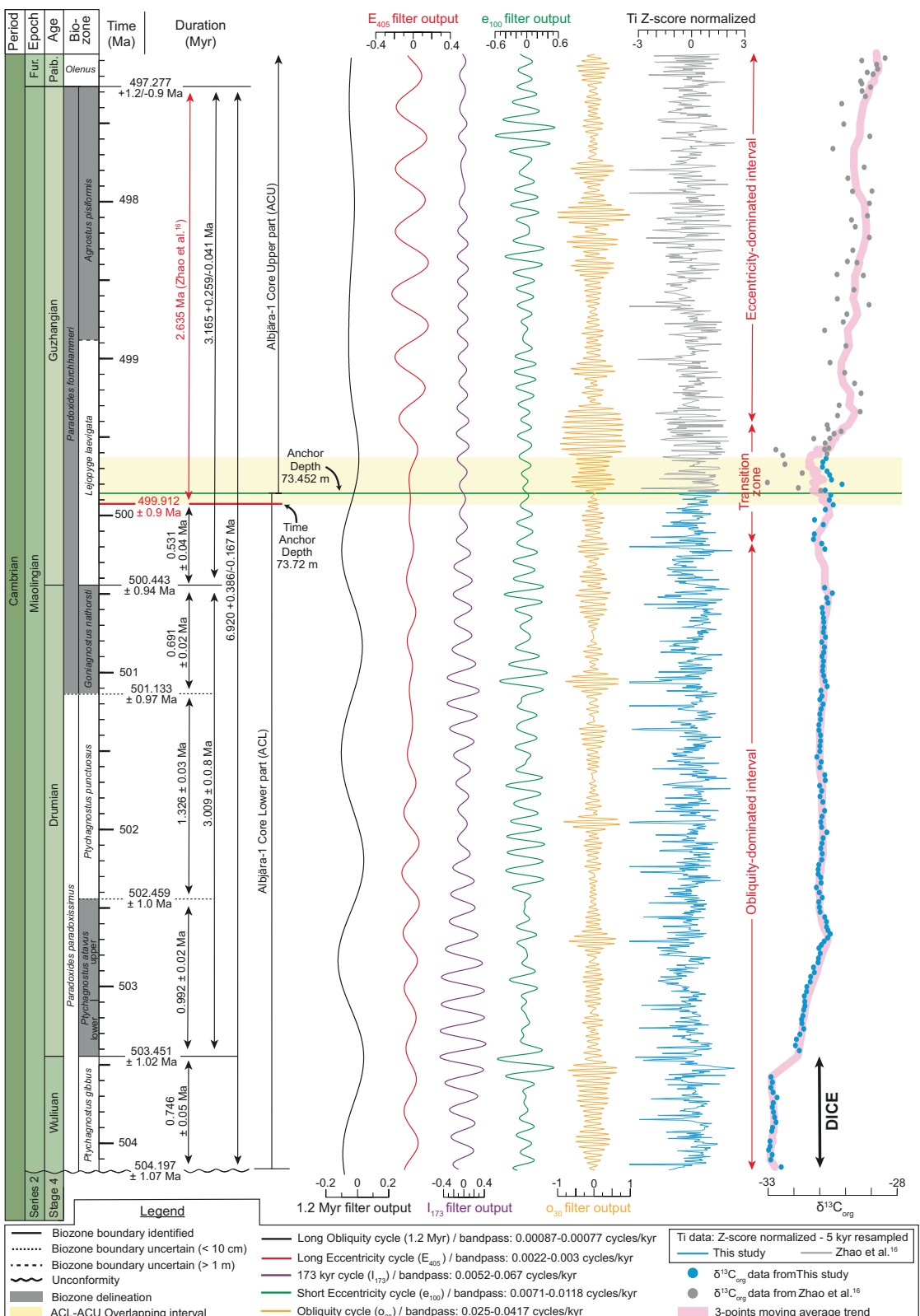

**Fig. 7 | Astronomically-calibrated bio- and chemostratigraphic framework of the Miaolingian Epoch in Baltica.** Filter outputs of the Milankovitch cycles, and 5 kyr-resampled tuned Titanium series (Z-score normalized). DICE Drumian Carbon isotope Excursion, $E_{405}$ Long Eccentricity, $e_{100}$ Short Eccentricity, Fur. Furongian, $I_{173}$ Inclination metronome, kyr kilo year, Ma Mega annum, Myr Million year, $o_{30}$ Obliquity, org organic, $p_{18}$ Precession, Paib. Paibian.

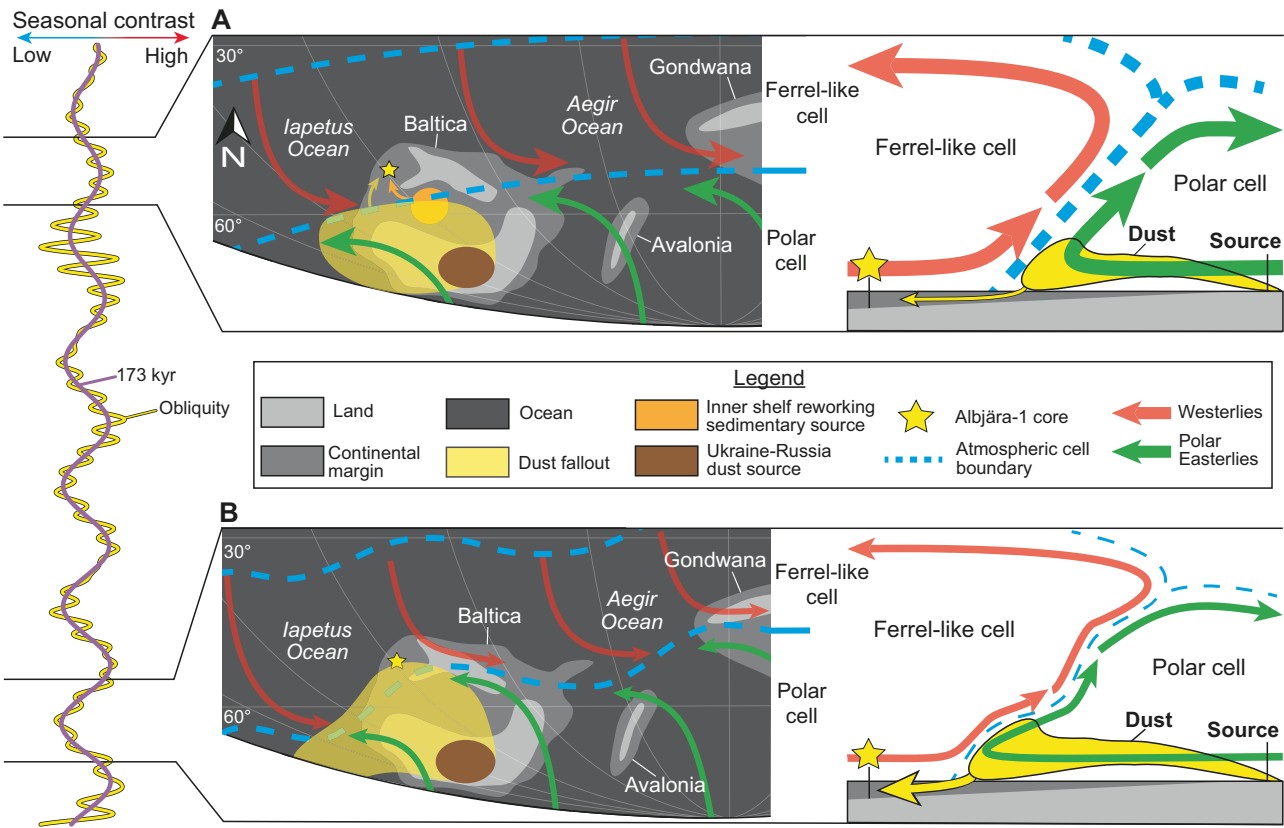

**Fig. 8 | Paleoclimatic source to sink model related to obliquity variation during obliquity-dominated time intervals in Baltica and relative sea-level curve.** **A** Obliquity maxima characterized by strong seasonal contrast resulting in a well-defined boundary between the Polar and Ferrel-like cells. Limited transport of dust from the source to Scania. **B** Obliquity minima characterized by weak seasonal contrast resulting in a disturbed boundary between Polar and Ferrel-like cells. Increased transport of dust from the source to Scania. Paleogeographic reconstruction modified from Jamart, et al.[4] with adjustment of the position of Avalonia and Gondwana based on Landing et al.[11], Keppie et al.[12], and Park[13].

Cambrian, the Alum Shale clastic source is thought to have been located in Russia-Ukraine, implying a source-to-sink distance of ≥ 1000 km[34]. Consequently, part of the variations of input into the Alum Shale of Scania can plausibly be attributed to obliquity-paced aeolian fluxes[17,34].

To understand the mechanism of aeolian dust delivery, it is necessary to consider atmospheric circulation, which is driven by latitudinal temperature gradients and solar forcing[71]. Today, the Hadley, Ferrel and Polar cells govern global climatic circulation and precipitation belts[72,73]. Cambrian climate reconstructions suggest three broad zones: 1) equatorial regions with maximum precipitation, 2) arid subtropics, and 3) cool to temperate conditions between the tropics and the poles with enhanced precipitation at 45–60° latitude[74,75]. Based on these models, the Hadley and Polar cells were likely already established with a latitudinal extents similar to that of today, separated by a Ferrel-like cell, placing Baltica (~60°S) across the boundary between the Ferrel-like and Polar cells (Fig. 8).

At obliquity maxima, strong seasonality produced cold winters and hot summers characteristic of a cold poles state[76–78]. In this state, the Ferrel-like–Polar boundary was strong and relatively stable due to steep temperature gradients, which limited dust transport across cells and reduced fallout in Scania (Fig. 8A). In contrast, obliquity minima produced mild summers and warm winters, characteristic of a warm poles state[76–78]. Under these conditions, the boundary between the Polar and Ferrel-like cells was weaker and more variable, allowing Polar Easterlies and Ferrel-like Westerlies to penetrate farther north and south, respectively, thereby enhancing dust transport to Scania (Fig. 8B).

To identify aeolian dust, the Ti/Al or Si/Al ratios are commonly used, with higher ratios indicating drier climates and enhanced dust fluxes, and lower ratios indicating wetter climates and reduced dust fluxes[79,80].

For obliquity-dominated intervals (e.g., latest Wuliuan-earliest Drumian, 502.4-503.5 Ma), at > 100 kyr scales, the dust ratios are in antiphase with both the Ti record and $I_{173}$ cycle, indicating that aeolian dust fluxes were the primary control on Ti delivery (Fig. 9A). At <100 kyr scales, during $o_{30}$ maxima, Ti maxima are in phase with $o_{30}$, while dust ratios remain antiphase to Ti, again suggesting a dominant dust contribution. Nonetheless, in a few cases, Ti and dust ratios are in phase, implying an additional control. This pattern likely reflects storm reworking of inner-shelf sediments (see the next section). Furthermore, intervals of $I_{173}$ minima (i.e., increased dust supply) coincide with $o_{30}$ minima (reduced hinterland erosion), thereby amplifying the dust signal in the Ti record.

For eccentricity-dominated intervals (e.g., latest Guzhangian – earliest Paibian, 497.1-498.2 Ma), Ti records show only weak $I_{173}$ modulation at > 100 kyr scales, and no consistent phase relationship is evident among Ti, $I_{173}$, and the dust ratios (Fig. 9B). Instead, Ti variability is primarily modulated by the $E_{405}$ and $e_{100}$ cycles. At <100 kyr scales, during $o_{30}$ maxima, Ti maxima are commonly in phase with the dust ratios, suggesting that sediment supply was dominated by erosion of the hinterland and/or storm-related reworking of inner-shelf sediments. In contrast to the obliquity-dominated interval, aeolian dust contributed only a minor component during this phase, with erosion and reworking processes acting as the main sedimentary source.

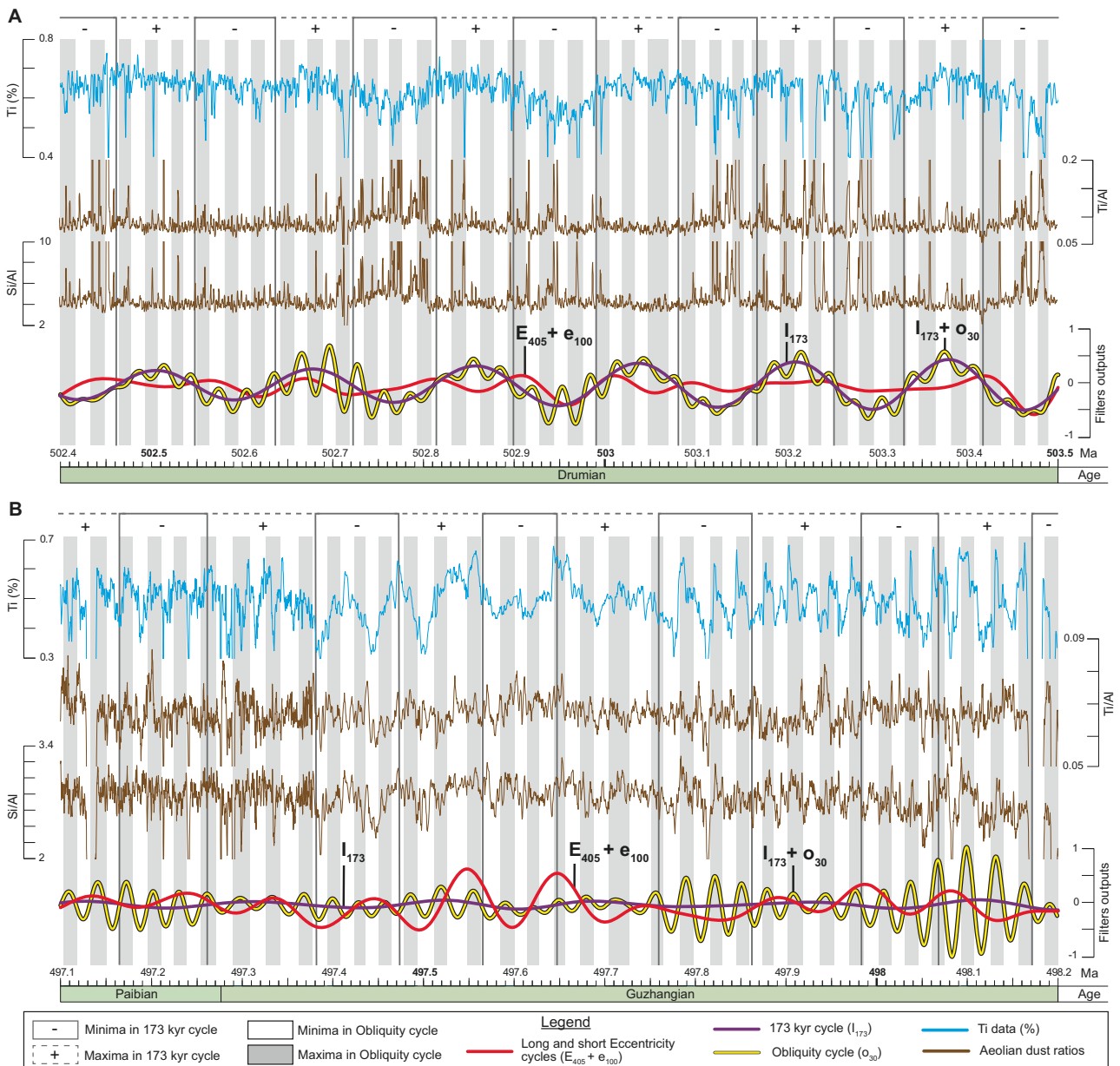

**Fig. 9 | Aeolian dust fluxes variations and Milankovitch-driven forcing.** Phase relationship between the Ti signal (%), aeolian dust ratios, and Milankovitch cycles for selected parts of the eccentricity-dominated and obliquity-dominated intervals. **A** Close-up view of the antiphase relationship between Ti (%), 173 kyr filter (obliquity-related cycle impacting the signal the most) and the aeolian dust ratios.

**B** Close-up view of the phase relationship between Ti (%), obliquity filter (obliquity-related cycle impacting the most the signal) and the aeolian dust ratios. The Ti (%), Ti/Al, and Si/Al curves correspond to the 5-points moving average trends observed in the data. $E_{405}$ Long Eccentricity cycle, $e_{100}$ Short Eccentricity cycle, $I_{173}$ 173 kyr cycle, kyr kilo year, Ma Mega annum, $o_{30}$ Obliquity cycle.

## Storms and reworking processes as drivers of Guzhangian detrital Input

The Scandinavian Shelf was exceptionally flat (slope of 0.1-1 m per km) and tectonically quiescent, which rendered it highly sensitive to eustatic fluctuations[33,34]. Even minor sea-level changes could impact the sedimentary record, explaining, for example, the extensive development of the Hawke Bay unconformity *sensu lato* in Baltica[14].

In addition to aeolian dust fluxes, Cambrian sediment supply to Scania was strongly influenced by reworking during sea-level lowstands, when a lowered storm wave base enabled repeated storms to remobilize shelf sediments along the Baltica margin and transport them into the deeper part of the Scandinavian Shelf (e.g., Scania)[14,16–18].

During the Guzhangian, the long eccentricity signal becomes stronger, while the $I_{173}$ modulation is weak to absent in the Ti record.

As shown in the previous section, the observed phase relationship between the aeolian dust ratios, Ti, and $o_{30}$ during eccentricity-dominated intervals indicates that aeolian dust was no longer the dominant detrital source (Fig. 9B). We suggest instead that the reworking of the shelf sediments, driven by minor sea-level changes and repeated storm events[14,18] became the primary mechanism of detrital supply.

## Lag-1 autocorrelation and eustatic variations

To test our paleoclimatic model, we performed a lag-1 autocorrelation ($\rho_1$) analysis (see Supplementary Information and Supplementary Code) on the tuned Ti series. The lag-1 autocorrelation is an independent statistical testing that serves as a robust indicator of sea-level variations: high $\rho_1$ values correspond to high sea-level and low $\rho_1$ values

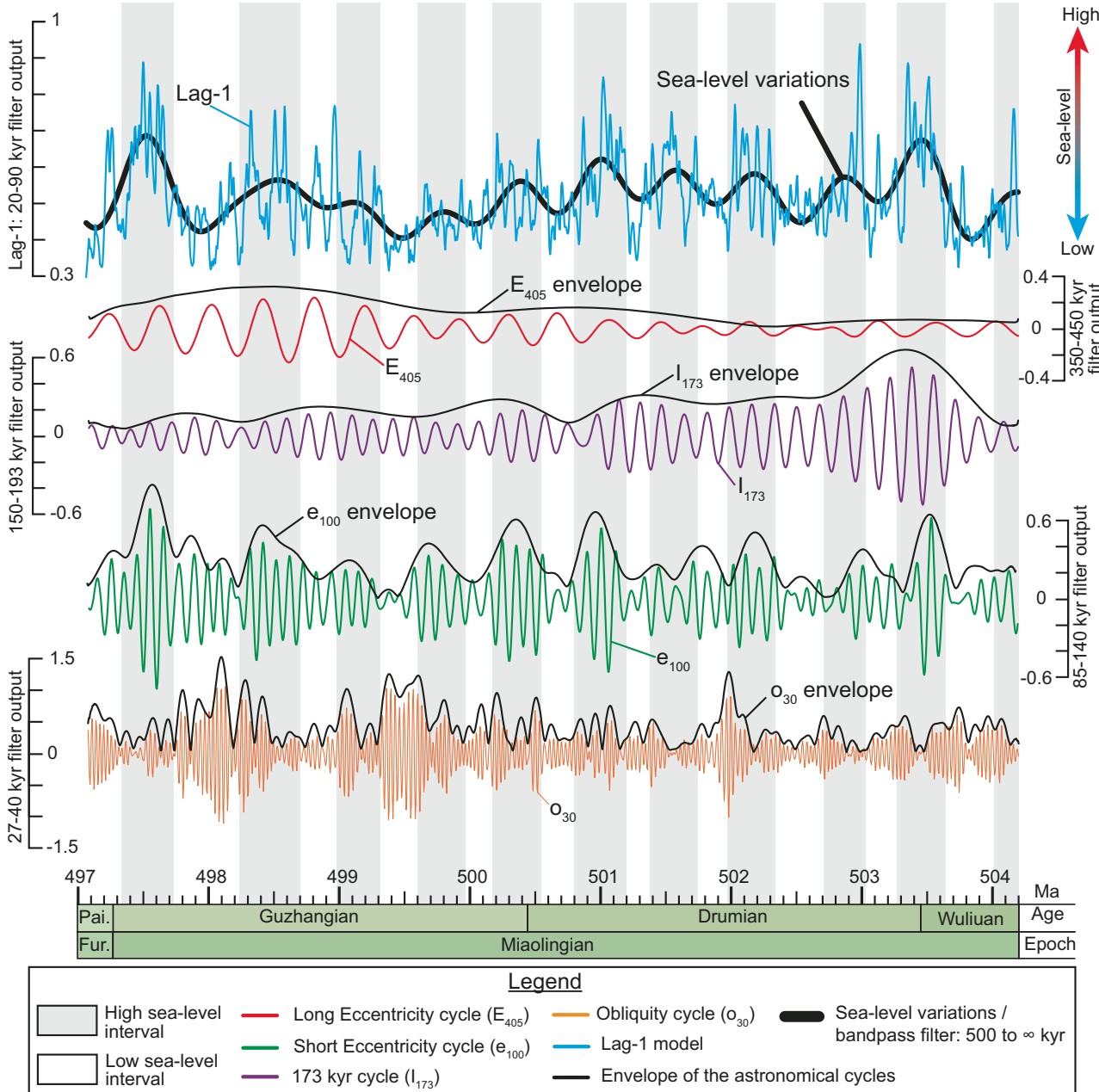

**Fig. 10 | Sea-level variations and Milankovitch-driven forcing.** Phase relationship between the lag-1 autocorrelation analysis (data resampled every 5 kyr) and the Milankovitch periodicities during the Miaolingian Series. $E_{405}$ Long Eccentricity cycle, $e_{100}$ Short Eccentricity cycle; Fur. Furongian, $I_{173}$ 173 kyr cycle, kyr kilo year, Ma Mega annum, $o_{30}$ Obliquity cycle, Pai. Paibian.

to low sea-level (Fig. 10)[81,82]. We compared the $\rho_1$ values with the filtered Milankovitch periodicities and their Hilbert envelopes (Fig. 10). This comparison reveals that the peaks in the eccentricity (essentially the $e_{100}$) are generally in phase with the $\rho_1$ values, whereas the peaks of the $I_{173}$ are in antiphase with these values (Fig. 10).

This suggests that the eccentricity exerted a strong influence on short-term sea level variations driven by aquifer-eustasy[83], while the antiphase behavior of obliquity implies that glacio-eustasy was not the controlling mechanism[84]. This interpretation aligns with the view of the Cambrian as an ice-free, greenhouse-to-hothouse period[2,23,84]. Assessing the Hilbert envelopes further highlights that maxima in short eccentricity align with high sea-level, while maxima in $o_{30}$ coincide with low sea-level (Fig. 10).

According to Sames et al.[83], in greenhouse conditions, the aquifer-eustasy mechanism operates as follows: during warm and humid

periods, the aquifers fill up, lowering the sea-level; during arid periods, the aquifers empty out, increasing the sea-level. In the present case study, we propose that during $e_{100}$ maxima, arid periods are intensified due to an increase in the Earth's solar energetic budget, leading to evaporation exceeding precipitation. This causes the aquifer to discharge and elevates sea-level. Conversely, during $e_{100}$ minima, arid periods are less pronounced, leading to precipitation exceeding evaporation and favoring aquifer recharge, thereby lowering sea-level.

The implications for our paleoclimatic model are as follows:

On the one hand, in the obliquity-dominated interval, maxima in $\rho_1$ (high sea-level) coincide with $I_{173}$ minima and high aeolian dust ratios. This indicates that when the sea-level was high, the storm wave base was elevated, reducing shelf reworking, and aeolian dust became the main source of Ti. On the contrary, during low sea-levels, $I_{173}$ maxima coincide with low aeolian dust ratios reflecting reduced dust

input (due to a strengthened Ferrel-like–Polar cells boundary) and enhanced reworking of inner shelf sediments as the storm wave base shifted downward.

On the other hand, in the transition to eccentricity-dominated intervals, the imprint of $I_{173}$ weakens and eventually disappears, while eccentricity cycles become the main modulator of the Ti record. The persistent phase relationship between $\rho_1$ and $e_{100}$ suggests that during these intervals, reworking of inner-shelf sediments was the dominant mechanism of Ti supply.

## Methods

We studied approximately two meters above the ACL to allow for correlation with the previously published studies on the ACU. Most of the ACL (212.2-232.62 m) was analyzed using XRF core-scanning, ICP-MS geochemistry, Rock-Eval pyrolysis, organic carbon isotope analysis, XRD mineralogy identification, fossil identification, and advanced time-series analysis. The lowermost interval (232.62-237.4 m), corresponding to the Gislöv and Hardeberga formations, was too lithified to allow effective sampling with hand drills; therefore, only XRF core-scanning was conducted in this interval. In this manuscript, the term ACL thus refers exclusively to the 20 m interval from 212.2 to 232.62 m, where multi-proxy analysis was successfully performed. All the fossil and geologic samples presented in this study are from the Albjära-1 core, curated at the University of Copenhagen in Denmark.

### Core sampling and sample preparation

A total of 151 powdered samples were collected at 15 cm resolution across the ACL. Sampling was carried out on carefully cleaned rock surfaces free of calcite veins and macroscopic pyrite, using an electric drill with a tungsten masonry drill bit. Between 5 and 10 g of rock powder were extracted per sample. These powders were used for organic geochemistry, XRD, and ICP-MS analyses.

### X-ray fluorescence (XRF) core-scanning

The ACL was scanned with an Itrax core scanner (University of Stockholm, Sweden) using the analytical conditions detailed in Supplementary Information. Measured concentrations of major (Al, Si, K, Ca, Ti, Fe in wt.% %) and trace elements (V, Cr, Mn, Ni, Cu, Zn, Ga, As, Rb, Sr, Y, Zr, Ba, Pb, U in ppm) are listed in Supplementary datasets.

### Inductively coupled plasma mass spectrometry (ICP-MS)

To calibrate the XRF core-scanning results, 50 samples were analyzed for major and trace element concentrations at Actlabs Activation Laboratories Ltd (Ancaster, Ontario, Canada). The samples were prepared following the Ultratrace 4 – near-total digestion method and analyzed by inductively coupled plasma mass spectrometry (ICP-MS; see analytical conditions in the Supplementary Information).

### Rock-eval pyrolysis

The 151 samples were analyzed with a Rock Eval 6 instrument (Technologies Vinci, Rueil-Malmaison, France) using the IFP 160 000 standard at the Institute of Earth Sciences of the University of Lausanne (ISTE, UNIL). 60 to 70 mg of powdered bulk rock were placed in an oven, heated at 300 °C under an inert atmosphere, and then gradually pyrolyzed up to 650 °C. After pyrolysis, the samples were transferred to another oven and heated to 850 °C in air, with $CO_2$ and hydrocarbon (HC) concentrations monitored throughout. The measured parameters included the total organic carbon content (TOC in wt %), carbonate content ($CaCO_3$, wt %), hydrogen index (HI, mg HC/g TOC), oxygen index (OI, mg CO2/g TOC), S2 peak (mg HC/g), and T max (°C). The analytic error for the TOC measurement is of 0.1 wt %[85].

### Carbon stable isotopes ($\delta^{13}C_{org}$)

The organic carbon stable isotope compositions ($\delta^{13}C_{org}$ values, ‰ vs. VPDB) were determined for all 151 decarbonated samples with TOC

contents >0.1 wt% using elemental analysis and isotope-ratio mass spectrometry. The used EA/IRMS system at the Institute of Earth Surface Dynamics of the University of Lausanne (UNIL, IDYST) consisted of a Carlo Erba 1108 (Fisons Instruments, Milan, Italy) elemental analyzer connected to a Delta V Plus isotope-ratio mass spectrometer via Con-Flo III split interface (both of Thermo Fisher Scientific, Bremen, Germany) operated under continuous helium flow. Before analysis, the samples were decarbonated by treatment with 10% v/v HCl, thoroughly washed with deionized water, and dried (40 °C, 48 h). The calibration and normalization of the measured $^{13}C$ values to the VPDB scale was performed with international reference materials and UNIL in-house standards[86,87]. The repeatability and intermediate precision were better than 0.1‰ for $\delta^{13}C_{org}$

### X-ray diffraction (XRD)

The composition of the clay minerals throughout the ACL was determined in 8 decarbonated core samples, using an X-TRA Thermo-ARL XRD system at the Institute of Earth Sciences (ISTE), University of Lausanne. The XRD patterns were acquired on 800 mg powder samples pressed (20 bars) into a powder holder (energy, 40 kV, 45 mA; CuKα, λ = 1.54060 Å; angle 2°–5° 2°θ) to determine the mineralogical composition of the bulk rock against external standards. A margin of error of 5-10% was permitted for phyllosilicates and 5% for grain minerals. Diffractograms were used to better identify minerals by peak position (2θ)[88], in addition to semi-quantitative approach used for clay minerals quantification

### Biostratigraphy

To improve the biostratigraphic resolution, a total of 230 fossils (agnostoids, trilobites, and brachiopods) exposed on core surfaces were photographed using a Samsung Galaxy A52 (smartphone) camera with a binocular lens and subsequently determined to the lowest possible taxonomic level.

### Cyclostratigraphy and time series analysis

To identify astronomical forcing in the sedimentary record, detrital elements (Al, Si, Ti, K, Zr), major constituents of clay minerals, are commonly used because their fluctuations are interpreted as direct responses to astronomically-forced climatic variations[16,17,25,26,78]. The usefulness of these elements also reflects their relative inertness during diagenesis[27,80]. However, more caution is required regarding K due to the illitization process occurring through burial diagenesis, especially in shales, which can significantly influence the K signal[80].

To align our dataset with that provided by Zhao et al.[16], we compared concentrations of detrital elements (Si, Al, Ti) and Ca in the -2 m overlapping interval between ACL and ACU. This comparison allowed us to identify a distinctive 5 cm calcite-vein framework at 212.22 m, which is clearly expressed in the XRF core-scanning data. This alignment serves as the reference point to construct our depth model.

Although XRF core-scanning was carried out on the interval 232.62–237.4 m, this section was not studied owing to the presence of a major unconformity between 232.616 and 232.745 m, which disrupts the stratigraphic continuity and prevents reliable time-series interpretation.

Prior to the time series analysis, the data were prepared by 1) reducing the thickness of anthraconite (diagenetic limestone) to 20%, 2) normalizing the data using Z-score normalization, and 3) calculating the theoretical durations of the targeted Milankovitch cycles (see details in Supplementary Information). After reducing the anthraconite thickness, we anchored our data at 212.22 m, corresponding to an adjusted depth of 73.452 m in Zhao et al.[16].

Detection and interpretation of Milankovitch cycles in the ACL were then undertaken using the Astrochron and WaverideR R packages[22,28]. The complete protocol for the spectral analysis is

provided in Supplementary Information and Supplementary Code (R scripts).

To assess the reliability and applicability for astronomical tuning of the Ti series on the ACL, we applied Average Spectral Misfit (ASM), Multi-taper Method (MTM), Evolutive Harmonic Analysis (EHA), and the frequency ratio methods (see Supplementary Information for more details).

### Time series protocol

The following section provides the spectral analysis protocol for this study. For more details for each of these steps, refer to Supplementary Information.

The protocol is as follows:

i. The data were resampled at a 5 mm resolution to ensure a constant spacing between sample points and to limit high-frequency noise. Then, a 20% locally weighted scatterplot smoothing (LOWESS) trend was subtracted to avoid low-frequency noise, similar to Zhao et al.[16]. and Sørensen et al.[17].

ii. The Multi-Taper Method (MTM)[89] was performed using two Slepian tapers, which allowed for the identification of significant periodicities that are > 90% confidence level (90% CL) in the signal.

iii. The frequency ratios for periodicities > 90% CL were then compared to the theoretical ratios of Cambrian Milankovitch cycles, as reported in the literature.

iv. Evolutive Harmonic Analysis (EHA)[89] and Continuous Wavelet Transform (CWT)[90] were used to track variations in periodicities and power over time, which are suggestive of changes in sedimentation rate.

v. To estimate the sedimentation rate in the ACL, the Average Spectral Misfit method (ASM)[91] was performed by integrating the > 90 % CL frequencies with regard to their theoretical duration.

vi. To better visualize the imprint of the > 90% CL periodicities in the sedimentary record, they were accurately isolated in the stratigraphic domain ( = depth domain) using a Taner bandpass filtering[92].

vii. We passed from depth to time by tuning our dataset using the trough function (astrochron R package) on the minima of the dominant periodicity (0.76-1 m), which we assumed to correspond to the 173 kyr cycle ( = Inclination metronome).

viii. The analyses presented in points **ii** and **iv** were applied to the floating astronomically-tuned dataset in the time domain.

ix. To test the phase relationship between obliquity band frequencies and the 173 kyr cycle in the time domain, we performed an amplitude modulation analysis. First, we filtered out the obliquity band using a 28–40 kyr bandpass Taner filter. Then we isolated the 173 kyr cycle using a 155–195 kyr bandpass Taner filter from the Hilbert transform of the obliquity. We extracted the 173 kyr cycle from the global signal using a 155–195-kyr Taner bandpass filter. Finally, we compared the 173 kyr cycle extracted from the dataset with the obliquity-band-extracted 173 kyr cycle.

x. Step **ix** was replicated to test the phase relationship between the 173 kyr cycle (155–195 kyr bandpass) and the long obliquity modulation (1150–1300 kyr bandpass).

xi. By tracking the 173 kyr cycle in CWT using WaverideR for the Al, Si, Ti, and K elements, we built an age-depth model with a maximum cumulative uncertainty of ±167 kyr over 4.3 Myr. This also enabled us to reduce the uncertainties of biozone durations to less than 10%.

xii. The high-resolution floating astronomical time scale obtained is anchored at 73.72 m AD to the absolute age of 499.9 ± 0.9 Ma reported by Zhao et al.[16].

xiii. To identify the primary orbital forcing in the sedimentary record of Baltica, the composite ATS spanning the entire Miaolingian Series was filtered using Milankovitch frequencies in the time domain.

xiv. Lag-1 analysis was performed on the Ti series to identify the relationship between Milankovitch cycles and sea-level fluctuations.

## Data availability

Supplementary datasets, which contain the data generated in this study, have been deposited in the Zenodo database under the accession link:https://doi.org/10.5281/zenodo.18466190. Supplementary Information, which contains additional information related to the main text, Supplementary Figs. as well as the detailed protocol used for the time series analysis, is available in the supplementary files associated with this work.

## Code availability

The R code related to the spectral analysis has been uploaded on GitHub repository under the accession link: https://github.com/ValentinJamart/Supplementary_Code_Albjara-1_core.git.

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

## Acknowledgements

This work is supported by an Ambizione fellowship awarded by the Swiss National Science Foundation to D.P. (grant n°193520). L.A.H. was generously supported by the Heising-Simons Foundation (Grant No. 2021-2796). We are grateful to Thomas Weidner, Denmark, for his help in the agnostoid identification.

## Author contributions

V.J. and D.P. designed the research. A.T.N. provided logistical support to access and work on the core material. V.J. collected and analyzed the data, prepared the figures, and wrote the initial version of the manuscript under the supervision of D.P.; L.A.H. provided support and supervision during a research stay in her laboratory. M.A. provided support for coding. V.J. wrote the manuscript with input from D.P., L.A.H., J.E.S., T.A., A.T.N., N.H.S., N.T., M.A., and A.C.D.

## Competing interests

The authors declare no competing interests.
