## [Transparent Peer Review file · Nature Communications]

Astronomical calibration of the middle Cambrian in Baltica: Global carbon cycle synchronization and climate dynamics

Corresponding Author: Mr Valentin Jamart

Version 0:

Reviewer comments:

Reviewer #1

(Remarks to the Author)

The authors provide new datasets of 15 cm-resolution $\delta^{13}\text{C}_{\text{org}}$, and a 1 mm-resolution X-ray fluorescence (XRF) core-scanning across the upper Wuliuan – lower Guzhangian succession in the Albjära-1 core. They improve the existing biostratigraphic scheme and combine this to refine location of top and base of Drumian and the location of the DICE carbon isotope excursion in the Alum Shale and to develop an astronomical age model for this core. Further a new paleoclimatic model that links sedimentary delivery in Baltica to Milankovitch forcing during the Miaolingian is provided. Overall, this is a well written paper providing valuable new data. Below a series of remarks, suggestions and questions

The challenge in this type of studies lies partly in the rock record. In line 194 the authors state “and stratigraphically continuous Albjära-1 core”, but how do we really know? I think it is valid to argue that the outer shelf depositional environment indeed is a quite environment that has the potential to record continuous sedimentation, but how do the limestone intervals interfere with this? And how complete is the core? Do you have full core recovery? Are parts missing? How are you dealing with core breaks? Is it one core or some sort of composite?

Line 141 & 470, supplement line 179, line 278 and several other places: the U-Pb age of 499.9115 ± 0.9 Ma age has too many decimals if uncertainty of 0.9 Ma is reported. Zhao et al. 2022 report 499.9 ± 0.9 Ma in their fig 4.

Line 767: “To improve the quality of biostratigraphic correlations with the global Cambrian biostratigraphic framework, a total of 230 fossils present in core cracks.” Only 18 photos are shown in appendix and only 62 are reported in the supporting Excel file. Where are the remaining data? And if fossils are only present in core cracks, is there any concern about mixing/reworking during the drilling process? And how much does that add to the uncertainty

Line 602 “For obliquity-dominated intervals (e.g., latest Wuliuan-earliest Drumian, 502.4-503.5 602 Ma), at > 100 kyr scales, the dust ratios are in antiphase with both the Ti record and I173 cycle, indicating that aeolian dust fluxes were the primary control on Ti delivery (Fig. 8A).”

When I look at 8A the Ti/Al and Si/Al ratios are high in both min and max of 173kyr cycle (503.1-503.2 Ma), so I do not fully understand what the authors mean.

Line 549 “and/or the northward drift of Baltica, which carried the craton farther into the Ferrel-like atmospheric cell, reducing aeolian dust transport to Scania and weakening the obliquity imprint.” So then I'll assume no obliquity in younger part of core as reported in Zhao et al., 2022, because Baltica is moving to equator. This might support this suggestion.

Line 682 “ This suggests that the eccentricity exerted a strong influence on short-term sea-level variations driven by aquifer-eustasy”. Briefly explain mechanism, why is sea level high in short eccentricity maximum?

Data sets and supplements seem complete to me.

In general: there are a number of typos, incorrect use of capital letters, different font sizes etc. in the supplement. I recommend that the authors carefully re-read their supplement again and correct this (e.g. line 772 excepted = expected, but there are more).

I understand why the authors do the Z score normalisation, but what about the sedimentation rate of the limestone versus the shale. Is that also on the same order?

Figure S2: how can I link the photos to stratigraphic position in core?

Line 396 "In the Albjära-1 core, the lowest occurrence of *P. atavus* is identified at ~90.295 m allowing the delineation of the Wuliuan – Drumian boundary in the ACL (see section 2.1.1. Fossils identification)." In the Excel supplement this is at 89.695m adjusted depth. *T. fissus* is recorded at that level, although *T. fissus* is not reported in the Excel table in the supplement. This should be added.

When discussing DICE maybe refer to <https://doi.org/10.1016/j.sedgeo.2025.106875>

Line 506-509: another argument that Ti was not mobile could be the lack of authigenic titania minerals. Can authors confirm that those were not present?

Figure S8: ratios of frequencies: in fact this is ratio of inverse of frequency

Line 782: this is fitting of frequencies / counting of cycles, not looking patterns that correspond with an astronomical solution, so suggested uncertainty is really small, but is it realistic? If a series of cycles is lacking, this would not necessarily be accounted for in this method. Independent checks with radio isotope data are also not available for this interval.

Line 830 Fig 6C you mean fig 5C? In fig 6 I also would like to see the filter output of the 1150-1300kyr filter plotted. When I look at the filtered records of O30 and I173 in figure 6 I would expect to see this phase relation in the lower part (line 833), but I do not find this very obvious from the filtered records

Line 853 "From 497.085 to ~499.4 Ma, the signal appears to be dominantly modulated by long and short eccentricity cycles rather than by obliquity which is present but does not dominate the signal (Fig. S13A)" To me fig S13 is rather crucial because here the more visual approach of astrochronology and the more statistical approach are integrated and I do not find it as straightforward as suggested by the authors. If I look at the interval shown in figure S13 the eccentricity does not always line up with the data: maxima seems to correspond with maxima in the normalized Z score, but for example at 497.45Ma the 100e max corresponds to a minimum in the Z score. In the interval 497.8 -498.2 Ma the obliquity seems to dominate, not eccentricity. Admittedly, the relation between the proxy record and filter outputs is also shown in main text in figure 6, but because this is the full record shown in one figure more detailed checks of agreement between filters and original data is somewhat difficult.

Line 901 2.4.1 What do the uncertainties mentioned here exactly incorporate? Also the uncertainty of the exact position of a bio-event?

Line 41: positioned positions

Line 81-84: check grammar

Supplement line 179: "To anchor our astronomical time scale (ATS), we used the absolute age of 499.9115 ± 0.9 Ma" replace "absolute age" with "astronomical age".

Line 160-161: "During the Miaolingian, the Albjära-1 drill site in Scania was located on the western margin of Baltica, at high southern latitudes (Fig. 1)" based on figure 1 drill core is at ~50 degrees south, high southern latitudes are >60 degrees

Line 186: "the green curve" these are dots/datapoints, not a curve

Line 194: "and stratigraphically continuous Albjära-1 core"

Line 204: "The possible negative DICE excursion" possibly negative, or possibly DICE?

Line 279-280 "This offset may reflect asynchrony in either the onset of the DICE or the first appearance of *P. Atavus*" Or both? Or issues with carbon isotope records? What is interpreted as DICE? Based on fig 3B there is wiggle room for interpretation

Supplement line 341 "identified by in the" rephrase

Zhao et al 2022 used different proxies for tuning the same core. Authors describe that they used Ti to facilitate integration with the dataset published by Zhao, et al. 2022 (line 323), but Zhao et al. use Al and Mo as main proxy. Would Al use yield an identical tuning result as Ti for the ACL? And vice versa, would ACU yield an identical result if Ti is used as main proxy instead of Al?

An overall remark: a lot of information is in the supplement. Reading this article requires a lot of going back and forth between main text and supplement. I do not find this very convenient, but also do not have an obvious solution for that.

(Remarks on code availability)

Reviewer #2

(Remarks to the Author)

To the editor and authors: on Jamart et al. ms.

Overall: This is thoroughly researched with the most up-to-date techniques with results recorded in text figs. and in supplementary texts. It will be a standard for comparison with “moderate” revision.” The title and abstract are appropriate, the text generally reads well (some line edits done), and text length and figures are all appropriate.

Problems: As in caption of Fig. 1, a problem that affects most figs. including supplementary and the text, is that the ages on base and top of “Miaolingian” are very off and should be 506.3 Ma and 494.5 (note references not cited). This does not seem to affect conclusions, but, in addition, the duration of the Drumian can be ca. 1 to 2 Ma longer (ca. 505 +/- 1 Ma base and top 500 +/- 1 Ma) and this means a problem for astrochronology that should be addressed.

Continuous deposition is assumed in this and preceding Zhou paper, this is difficult to envision when the environment includes intervals of heightened storminess. Needs discussion.

The “standard” interpretation of Alum Shales as “deep” should be muted as Alum muds deposited on shorelines (Landing & Geyer, 2025; Landing 2025 Bakken mud model).

The inclusion of Avalonia as an appendage on Gondwana is a problem that has to be addressed at least in the captions of several figures: Landing et al. (2022, 2023), Keppie, Keppie, and Landing (2023), and Park (2025) show Avalonia as separate ribbon continent close to Baltica.

A final “problem” is interpretation of Cambrian climates as warm even to the poles. At a time of no glaciation, existing studies on oxygen isotopes cannot provide a reasonable paleotemperature. Anyway, the paleogeog map used in the ms. has Baltica in relatively low latitudes during a very warm interval and somehow lacks significant carbonates? And has the same siliciclastic lithofacies as Avalonia shown far to the south?

The paper is frankly difficult to read for people not involved in astrochronology studies. An “Analytical Procedures” section that briefly (an added page) preceding the “Spectral analysis” section (line 430s) maybe should explain terminology like “inclination metronome,” “time domain analysis,” obliquity modulation,” etc.

Ed Landing as reviewer can be noted.

Landing, E. The Bakken Model: Deposition of Organic-Rich Mudstones and Petroleum Source Rocks as Shallow-Marine Facies Through the Phanerozoic. *J. Mar. Sci. Eng.* 2025, 13, 895. <https://doi.org/10.3390/jmse13050895>

Landing E, Schmitz MD, Westrop SR, and Geyer G (2023) U-Pb zircon dates from North American and British Avalonia bracket the Lower–Middle Cambrian boundary interval, with evaluation of the Miaolingian Series as a global unit. *Geological Magazine* 160: 1790–1816. <https://doi.org/10.1017/S0016756823000729>

Landing E, Geyer G (2025) Comment: Carbonate production and reef building under ferruginous seawater conditions in the Cambrian rift branches of the Avalon Zone, Newfoundland by J.J. Alvaro and A. Mills, *Sedimentology* (71, 1245–1269). *Sedimentology* doi: 10.1111/sed.13253

(Remarks on code availability)

Reviewer #3

(Remarks to the Author)

I'm sorry for the delay in providing my review comments. I accepted to review this manuscript because, from the abstract, I understood that it dealt with C chemostratigraphy. However, upon closer reading, I found that the manuscript covers many aspects, most of which are outside my domain of expertise. Therefore, I feel that I am not eligible to provide a robust and comprehensive review.

However, I see that the manuscript adopts a robust multidisciplinary approach to detect the DICE in a new locality, which provides important insights into the Cambrian period, including the possible non-synchronicity of the DICE and the occurrence of biostratigraphic index fossils such as *P. atavus*. From my domain of expertise, I find that the use of $\delta^{13}\text{C}_{\text{org}}$ to detect the DICE is not the most appropriate methodology, since $\delta^{13}\text{C}_{\text{org}}$ may not be synchronous with $\delta^{13}\text{C}_{\text{carb}}$. This is because $\delta^{13}\text{C}_{\text{org}}$ can be altered by the addition of detrital organic carbon, particularly given that the negative excursion identified by the authors occurs directly above an erosional unconformity, which may reflect detrital input.

A new methodology is being developed that allows the measurement of C_{org} from shale

(see paper " Bowyer, F., Yilales, M., Wood, R., & Poulton, S. W. (2023). Insights into the terminal Ediacaran marine carbonate record from shale-hosted carbonate carbon isotopes. American Journal of Science, 323.").

I believe that if C_{carb} is also measured in the section, it would provide more robust data for correlation with regional sections

Line 226: The authors refer to Figure 2 when describing the C_{org} variation with stratigraphy ('from 93.611 to 91.229 m position...'). However, this interval is not illustrated in Figure 2. I recommend homogenizing the metric scale in Figure 2, as the interval '93.611 to 91.229 m' cannot be located. This observation applies to the whole "Carbon stable isotopes ($\delta^{13}\text{C}_{\text{org}}$)"sections."

(Remarks on code availability)

Version 1:

Reviewer comments:

Reviewer #2

(Remarks to the Author)

To Valentin et al.,

This is simply one of the most thorough original astrochronological analyses within a temporally significant part of the Cambrian. I said as much in my original review. The discussion goes well beyond a single core, with the discussion including a quite "rational" reconstruction of possible weather circulation with a revised Cambrian paleogeography. The authors constructively added to the original ms. by thoroughly responding to the three reviewers, which is evident in the numerous text additions and changes in the revised ms.

The only "problem" is that their approach thoroughly transcends the "old fashioned" lithostrat and biostrat approach. Though, I can quibble that blankets of strongly dysoxic black mudstone really must be chock-full of minor diastems with more time represented by micro-unconformities than by the now compacted sedimentary rock. But, this does not affect their analysis in any way.

The ms. now has enough aids to help navigate the complex analysis. Thus, it will be an aid in teaching.

Best,

Ed L.

(Remarks on code availability)

Reviewer #1

The authors provide new datasets of 15 cm-resolution $\delta^{13}\text{C}_{\text{org}}$, and a 1 mm-resolution X-ray fluorescence (XRF) core-scanning across the upper Wuliuan – lower Guzhangian succession in the Albjära-1 core. They improve the existing biostratigraphic scheme and combine this to refine location of top and base of Drumian and the location of the DICE carbon isotope excursion in the Alum Shale and to develop an astronomical age model for this core. Further a new paleoclimatic model that links sedimentary delivery in Baltica to Milankovitch forcing during the Miaolingian is provided. Overall, this is a well written paper providing valuable new data. Below a series of remarks, suggestions and questions

The challenge in this type of studies lies partly in the rock record. In line 194 the authors state “and stratigraphically continuous Albjära-1 core”, but how do we really know? I think it is valid to argue that the outer shelf depositional environment indeed is a quite environment that has the potential to record continuous sedimentation, but how do the limestone intervals interfere with this? And how complete is the core? Do you have full core recovery? Are parts missing? How are you dealing with core breaks? Is it one core or some sort of composite?

The data come from a single core with an almost 100% recovery rate. All of the biozones identified in Scandinavia have been recognized, with possible limited and negligible gaps. This suggests that, despite the minimal erosional surfaces, the limestone did not significantly impact these outer shelf settings.

Prior to conducting XRF core scanning a chlorine-rich dough was applied to the cracks and core breaks to easily identify them. Based on the in-well measurements, it was determined that a total of approximately 5 to 10 centimeters were missing from the breaks in the core. This was accounted for by adding a gap in the supplementary Table S1 before conducting the time series analysis.

Line 141 & 470, supplement line 179, line 278 and several other places: the U-Pb age of 499.9115 ± 0.9 Ma age has too many decimals if uncertainty of 0.9 Ma is reported. Zhao et al. 2022 report 499.9 ± 0.9 Ma in their fig 4.

The age of 499.9115 was taken from the supplementary file of Zhao et al. (2022) and corresponds to the exact time of the anchor point (73.72 m adjusted depth). To avoid confusion, the age has been corrected to 499.9 ± 0.9 Ma in the figures and text.

Line 767: “ “To improve the quality of biostratigraphic correlations with the global Cambrian biostratigraphic framework, a total of 230 fossils present in core cracks.” Only 18 photos are shown in appendix and only 62 are reported in the supporting Excel file. Where are the remaining data? And if fossils are only present in core cracks, is there any concern about mixing/reworking during the drilling process? And how much does that add to the uncertainty

The fossils within the cracks are not reworked because the parts and counterparts are both embedded within the core. This has no effect on the uncertainty of the fossils' position. The "cracks" correspond to bedding surfaces incidentally exposed due to the partial splitting of the core during transportation or manipulation.

The word “cracks” has been changed to “exposed on core surfaces”.

The 62 specimens in the supplementary Table S2 correspond to the depths at which the fossils were found. Several fossils of the same species could be found within the same depth interval, which is why a total of 230 specimens were counted. To improve clarity between the main text and the supplementary Text S1 and Table S2, the number of identified specimens was added to the supplementary Table S2 at their corresponding depth. In addition, the specimens identified by Lauridsen (2000) and Zhao et al (2022) were also added to the supplementary Table S2.

Line 602 “For obliquity-dominated intervals (e.g., latest Wuliuan-earliest Drumian, 502.4-503.5 602 Ma), at > 100 kyr scales, the dust ratios are in antiphase with both the Ti record and I173 cycle, indicating that aeolian dust fluxes were the primary control on Ti delivery (Fig. 8A).”

When I look at 8A the Ti/Al and Si/Al ratios are high in both min and max of 173kyr cycle (503.1-503.2 Ma), so I do not fully understand what the authors mean.

It's true that maintaining straightforward relationships over long periods of time can be difficult. This is certainly due to the fact that we are interpreting natural behavior, which, by definition, does not follow any linear rules and can be chaotic. For the 502.4-503.5 Ma interval, you are correct that the proposed model does seem to not always react as theoretically developed in the text. However, this is limited to a small part of the studied interval. Nevertheless, most of the relationships along the core, as displayed in Fig. 8A, align with the proposed paleoclimatic explanation.

In the 502.98–503.17 Ma interval (minimum of 173 kyr), the Ti/Al and Si/Al ratios are high at both the minimum and maximum of obliquity (blue and non-blue δ^{30} rectangles), which is consistent with our proposed model. In the 503.17–503.25 Ma interval for instance (maxima of 173 kyr), the Ti/Al and Si/Al ratios are high, which seem to partly invalidate our model. However, high peak values are only observed when obliquity is minimal (non-blue δ^{30} rectangles). These δ^{30} minima correspond to a short period when the boundary of the atmospheric cells is slightly weaker. Therefore, the high dust ratio values observed between 503.17 and 503.25 Ma are likely to result from a more pronounced weakening of the atmospheric cell boundaries within a period of global strengthening of these boundaries (173 kyr maximum), which allowed more dust to reach Scania. Consequently, even if the Ti/Al and Si/Al ratios are high during this period, this does not invalidate our proposed model.

To better visualize this relationship, I modified figure Fig 8 (see below) by adding the yellow I173+ δ^{30} filter at 50% opacity on the black Ti and Si/Al curves. This adding of another level of color to this busy figure might be worthwhile. We are waiting for your (or the editor's) feedback to decide which version (original or new) is more suitable for the final manuscript.

Line 549 “and/or the northward drift of Baltica, which carried the craton farther into the Ferrel-like atmospheric cell, reducing aeolian dust transport to Scania and weakening the obliquity imprint.” So then I’ll assume no obliquity in younger part of core as reported in Zhao et al., 2022, because Baltica is moving to equator. This might support this suggestion.

In Baltica, the long obliquity (173 kyr) gradually weakens until it disappears completely in the middle Paibian Stage, as shown in the supplementary material of Zhao et al. (2022). However, the 30 kyr obliquity remains strong during the upper Cambrian period. The northward migration of Baltica appears to be the most likely cause of the 173 kyr cycle's disappearance, but we cannot rule out chaotic behavior in the 173–152 kyr alternation entirely. Therefore, we prefer to maintain these two hypotheses rather than selecting one over the other based on insufficient evidence.

Line 682 “This suggests that the eccentricity exerted a strong influence on short-term sea-level variations driven by aquifer-eustasy”. Briefly explain mechanism, why is sea level high in short eccentricity maximum?

There is no evidence of specific deposits that could be used to confidently support the identification of aquifer-eustasy, as opposed to glacio-eustasy, which can be identified by the presence or absence of glacial deposits. Nevertheless, the literature proposes that aquifer-eustasy acts similarly to glacio-eustasy in greenhouse conditions (e.g., Sames et al, 2020; Li et al, 2018).

The following sentence was added to the main text to clarify the role of aquifer-eustasy:

According to Sames et al.⁸², in greenhouse conditions, the aquifer-eustasy mechanism operates as follows: during warm and humid periods, the aquifers fill up, lowering the sea level; during arid periods, the aquifers empty out, increasing the sea level. In the present case study, we propose that during e100 maxima, arid periods are intensified due to an increase in the solar energetic budget of the Earth, resulting in evaporation exceeding precipitation. This causes the discharge of the aquifer and elevates the sea level. Conversely, during e100 minima, arid periods are less pronounced, resulting in precipitation exceeding evaporation and favoring aquifer recharge, which lowers the sea level.

In general: there are a number of typos, incorrect use of capital letters, different font sizes etc. in the supplement. I recommend that the authors carefully re-read their supplement again and correct this (e.g. line 772 excepted = expected, but there are more).

I apologize for this. The supplementary text has now been revised and corrected.

I understand why the authors do the Z score normalisation, but what about the sedimentation rate of the limestone versus the shale. Is that also on the same order?

Below are a few points addressing it:

- 1) Neither Sørensen et al. (2020) nor Zhao et al. (2022) mentioned or demonstrated significant variation in sedimentation rates due to the limestone interval.
- 2) Fig. 4 shows a visual comparison of the original data and the Z-score. In the limestone interval, the cycles are slightly longer than in the shales, though this difference is not significant.
- 3) A significant change in sedimentation rate would have a major impact on the overall observed cyclicities in the EHA and CWT graphs, but that is not the case here.

We believe that the slight difference in sedimentation rates within the limestone interval is negligible and does not impact our age model.

Figure S2: how can I link the photos to stratigraphic position in core?

I apologize for this confusion. The depths of the presented fossils in Figure S2 have been added in the figure caption.

Line 396 “In the Albjära-1 core, the lowest occurrence of *P. atavus* is identified at ~90.295 m allowing the delineation of the Wuliuan – Drumian boundary in the ACL (see section 2.1.1. Fossils identification).” In the Excel supplement this is at 89.695m adjusted depth. *T. fissus* is recorded at that level, although *T. fissus* is not reported in the Excel table in the supplement. This should be added.

The Excel file only contains fossils that were identified and photographed during this study. The base of *T. fissus* at 90.295 m was determined through a graphical correlation with Lauridsen's (2000) data from the same core. This base was added to the supplementary Table S2.

When discussing DICE maybe refer to <https://doi.org/10.1016/j.sedgeo.2025.106875>

Thank you for the suggestion. We have added this reference to the supplementary text when discussing the DICE.

Line 506-509: another argument that Ti was not mobile could be the lack of authigenic titania minerals. Can authors confirm that those were not present?

Unfortunately, it was not possible to make thin sections of the core, so it is impossible to be completely sure that there are no authigenic titania minerals present. However, the literature and the XRD analysis in this paper do not show the presence of such minerals. Consequently, we believe that titanium remains a robust element for cyclostratigraphic analysis in this study.

Figure S8: ratios of frequencies: in fact this is ratio of inverse of frequency

In the astrochronology community, the period of a cycle is expressed in meters (m), while its frequency is expressed in cycles per meter (cycles/m) or cycles per kyr (cycles/kyr) when the dataset is tuned and in the present manuscript we followed this consensus.

Line 782: this is fitting of frequencies / counting of cycles, not looking patterns that correspond with an astronomical solution, so suggested uncertainty is really small, but is it realistic? If a series of cycles is lacking, this would not necessarily be accounted for in this method. Independent checks with radio isotope data are also not available for this interval.

You are right; If radioisotope dating were available it would help identify possible gaps. Based on biostratigraphy we can confirm that the record is complete and that no biozones, or significant parts of those, are missing from the Albjära-1 core, even though few thousand years might be missing. The low uncertainty in the age model is the result of a combination of four elements: Al, Ti, Si, and K. While some cycles might be missing from one or more proxies for various reasons, it is unlikely that all of them would miss the same cycle. Therefore, combining the time series analyses of these four proxies is a robust approach that reduces uncertainty. The exact same method has been developed and used in the Silurian of the Carnic Alps by Arts et al. (2024). Additionally, visually inspecting the Z-score of the core revealed no sharp discontinuities that could indicate time loss. This suggests that only a few thousand years may be missing. The missing few thousand years are encompassed within the age model's uncertainty. If requested, an additional 173 kyr of uncertainty could be added to the model, but we believe that the age model remains valid without it. Currently, there is no consensus on how to report age uncertainties (Sinnesael et al., 2019).

Line 830 Fig 6C ◊ you mean fig 5C? In fig 6 I also would like to see the filter output of the 1150-1300kyr filter plotted. When I look at the filtered records of O30 and I173 in figure 6 I would expect to see this phase relation in the lower part (line 833), but I do not find this very obvious from the filtered records

You are right. It is Figure 5C, not Figure 6C. This has been corrected.

A 1,150–1,300 kyr filter has been added to Figure 6.

The phase relationship between o30 and I173 might not be apparent in Fig. 6 because the obliquity filter differs from the one used in Fig. 5C. Figure 6 filters the entire obliquity band (24–40 kyr), while Figure 5C only filters the band ranging from 27 to 40 kyr. This range corresponds to the ~32-39 range (s3-s6) that creates the I173 modulation (Laskar, 2020; Laskar et al., 2004; Wu et al., 2024).

The 1,200-kyr filter has been added to the supplementary Excel file in the "Filters (time) 1 kyr" folder. Within the same excel folder, a clear separation has been added to the 31 kyr filter between the entire obliquity band and the band that generates the 173 kyr modulation. Step 13.3 of the code has been updated to include the 1,200-kyr filtering and the above-mentioned clarification within obliquity filtering.

Line 853 "From 497.085 to ~499.4 Ma, the signal appears to be dominantly modulated by long and short eccentricity cycles rather than by obliquity which is present but does not dominate the signal (Fig. S13A)" To me fig S13 is rather crucial because here the more visual approach of astrochronology and the more statistical approach are integrated and I do not find it as straightforward as suggested by the authors. If I look at the interval shpwn in figure S13 the eccentricity does not always line up with the data: maxima seems to correspond with maxima in the normalized Z score, but for example at 497.45Ma the 100e max corresponds to a minimum in the Z score. In the interval 497.8 -498.2 Ma the obliquity seems to dominate, not eccentricity. Admittedly, the relation between the proxy record and filter outputs is also shown in main text in figure 6, but because this is the full record shown in one figure more detailed checks of agreement between filters and original data is somewhat difficult.

To address the issue raised above, Figure S13 has been placed in the main text (= Fig. 6). At 497.45 Ma, the Ti data shows an uncommon variation that may not be linked to astronomical cycles, as none of the filters fully explains this variation. Thus, it can be linked to a combination of cycles for which we did not explore the imprint on the signal. For example, a strong ~50 kyr cycle, likely resulting from the combination of the ~20 kyr precession and the ~ 31 kyr obliquity, is present in the MTM, which could explain why the e100 at 497.45 Ma display such variation. Nevertheless, for the remainder of this interval, the eccentricity dominates the long-term trend observed in the Ti record.

Line 901 2.4.1 What do the uncertainties mentioned here exactly incorporate? Also the uncertainty of the exact position of a bio-event?

The uncertainties presented in Section 2.4.1 correspond to those of the age model itself. These uncertainties are calculated as follows: (maximum age of the biozone – minimum age of the biozone) – duration of the biozone.

These uncertainties account for variations in identifying astronomical cycles and the possible absence of a few thousand years of data in the core. The age model is based on four proxies, making it robust and allowing for lower uncertainty. If the model relied on only one proxy, the uncertainty would be ± 0.173 Ma.

Line 41: positioned positions

The change has been completed.

Line 81-84: check grammar

The change has been completed.

Supplement line 179: “To anchor our astronomical time scale (ATS), we used the absolute age of 499.9115 ± 0.9 Ma” \diamond replace “absolute age” with “astronomical age”.

The change has been made.

Line 160-161: “During the Miaolingian, the Albjära-1 drill site in Scania was located on the western margin of Baltica, at high southern latitudes (Fig. 1)” \diamond based on figure 1 drill core is at ~50 degrees south, high southern latitudes are >60 degrees

The change has been completed.

Line 186: “the green curve” \diamond these are dots/datapoints, not a curve

The green shape is a moving average, which is usually referred to as a "curve" in literature. To avoid confusion, it has been renamed "the green trend."

Line 194: “and stratigraphically continuous Albjära-1 core”

The sentence was adjusted as follows:

The Albjära-1 core is exceptionally well-preserved, with a recovery rate approaching 100%, and stratigraphically continuous, with all biozones known from Scandinavia³³. This makes the core a new regional reference section of global relevance for the outer-shelf of the Miaolingian Series in Baltica.

Line 204: “The possible negative DICE excursion” \diamond possibly negative, or possibly DICE?

This has been changed to “a negative excursion, possibly the DICE”

Line 279-280 “This offset may reflect asynchrony in either the onset of the DICE or the first appearance of *P. Atavus*” ◊ Or both? Or issues with carbon isotope records? What is interpreted as DICE? Based on fig 3B there is wiggle room for interpretation

As you rightly mentioned, interpreting the DICE is challenging. By definition, it is a negative excursion near the base of *P. atavus*; however, its relationship with *P. atavus* is more complex than previously believed as shown in Figure 3B. Without additional information about the time constraints at the beginning of the Drumian Stage, it is impossible to discuss this topic with confidence.

We added "or both" after "*P. atavus*" in the main text.

Supplement line 341 “identified by in the” rephrase

The word “by” was removed from the sentence.

Zhao et al 2022 used different proxies for tuning the same core. Authors describe that they used Ti to facilitate integration with the dataset published by Zhao, et al. 2022 (line 323), but Zhao et al. use Al and Mo as main proxy. Would Al use yield an identical tuning result as Ti for the ACL? And vice versa, would ACU yield an identical result if Ti is used as main proxy instead of Al?

We used Ti to firmly anchor our study to the one published by Zhao et al. (2022). However, our age model is based on a combination of four proxies: Al, Ti, Si, and K. In the supplementary Table 4, we present the astronomical timescale with regard to Ti. The same result would be obtained if Al, Si, or K were used instead, as they all exhibit similar overall variations in the depth domain.

In their study, Zhao et al. (2022) used only Al to create their astronomical timescale. However, as in our study, Zhao et al. (2002) verified that the same cyclicity was also recorded in other detrital elements, such as Si and Ti. Therefore, no difference would be observed if they had used Ti rather than Al in their study, or vice versa.

In their study, Zhao et al. (2020) used Mo as a redox proxy. However, they did not use it to create their timescale. Unfortunately, the XRF source in our study is made of molybdenum, so it is not possible to reliably measure the concentration of this element.

An overall remark: a lot of information is in the supplement. Reading this article requires a lot of going back and forth between main text and supplement. I do not find this very convenient, but also do not have an obvious solution for that.

We agree with you completely. Although a lot of information is presented in the supplementary materials, we did our best to condense the most important information and include it in the main text. We apologize for the back-and-forth between the main text and the supplementary materials.

Reviewer #2

Overall: This is thoroughly researched with the most up-to-date techniques with results recorded in text figs. and in supplementary texts. It will be a standard for comparison with “moderate” revision.” The title and abstract are appropriate, the text generally reads well (some line edits done), and text length and figures are all appropriate.

Problems: As in caption of Fig. 1, a problem that affects most figs. including supplementary and the text, is that the ages on base and top of “Miaolingian” are very off and should be 506.3 Ma and 494.5 (note references not cited). This does not seem to affect conclusions, but, in addition, the duration of the Drumian can be ca. 1 to 2 Ma longer (ca. 505 +/- 1 Ma base and top 500 +/- 1 Ma) and this means a problem for astrochronology that should be addressed.

The ages have been modified according to the latest version of the International Chronostratigraphic Chart (ICC v2024-12), a globally accepted reference. In this version, the base of the Wuliuan Stage is set at ~506.5 Ma, while the base of the Guzhangian Stage remains at ~497 Ma.

In the Supplementary Text 1, we added additional information to explain our suspicion of a possible Pb loss affecting ages in North America, as well as our reasoning for maintaining the base of the Furongian Series at 497 Ma instead of lowering it to 494.5 Ma.

Using the ages and stage durations reported by Farrell et al. (2025), the base of the Paibian Stage is 494.5 Ma, the top of the Drumian Stage is ~497.8 Ma, and the base of the Drumian Stage is ~500.8 Ma. These estimates imply a Drumian duration of ~3 Myr, consistent with our model (3.0 ± 0.1 Myr). However, the absolute ages they assign to the Drumian boundaries (~500.8 Ma for the base and ~497.8 Ma for the top) are not consistent with published U–Pb constraints that bracket the Drumian Stage.

The following sentence was added to the main text:

Our age estimate of 503.45 ± 1.02 Ma and 500.44 ± 0.94 Ma, respectively for the base and the top of the Drumian Stage closely aligns with available U–Pb ages bracketing this Stage in West Gondwana (base between 503.14 ± 0.13 Ma in Landing et al.⁶³ and 505 ± 1 Ma in Palacios et al.⁶⁴; top slightly above 500.9 ± 0.9 Ma in Palacios et al.⁶⁴) as well as with the U–Pb age of 501.45 ± 0.10 Ma identified in the upper Drumian of Avalonia in Landing et al.⁶⁵.

The Drumian Stage is located between the Andrarum (*L. laevigata* Zone) and Exsulans (*P. gibbus* Zone) limestone beds in the Albjära-1 core. This indicates that, despite the minimal erosional surfaces at the base of the Andrarum limestone, there are no gaps in this specific black mudstone part of the core. Regarding the detrital elements and/or GR data, no sharp unconformities were identified, suggesting a continuous depositional record. Additionally, the estimated duration of the *G. nathorsti* zone in this study (691 kyr) closely aligns with the equivalent biozone (*G. nathorsti* + *L. armata*) in southern China, which is estimated at 676 kyr. Consequently, it is unlikely that 1 or 2 million years are missing from this part of the core.

Continuous deposition is assumed in this and preceding Zhou paper, this is difficult to envision when the environment includes intervals of heightened storminess. Needs discussion.

The study site was located on the outer shelf of Baltica established as located below the storm wave base (Nielsen & Schovsbo, 2015), so it was not affected by regular storm events. In the present study, after a careful inspection of the Albjära-1 core, no reworking was observed in the shale and only a negligible few centimeters (transition from shale to limestone) present slightly reworked sediment. In addition, solely a few mm-thick silt to very fine sand interval, interpreted as thin distal tempestites, are observed in the Albjära-1 core (Lauridsen 2000; Nielsen & Schovsbo 2011, 2015). Finally, storms are geologically instantaneous events that do not significantly affect deposition in outer shelf and deeper environments.

The “standard” interpretation of Alum Shales as “deep” should be muted as Alum muds deposited on shorelines (Landing & Geyer, 2025; Landing 2025 Bakken mud model).

Scania is located on the deepest part of the Scandinavian Shelf. It might indeed not be basinal as you suggest but it is far enough to be located below the storm wave base in the outer shelf.

The text has been adjusted replacing “black shale” by “black mudstone” to avoid any confusion with basinal black shales deposits.

The inclusion of Avalonia as an appendage on Gondwana is a problem that has to be addressed at least in the captions of several figures: Landing et al. (2022, 2023), Keppie, Keppie, and Landing (2023), and Park (2025) show Avalonia as separate ribbon continent close to Baltica.

Thank you for your comment. We have adjusted the paleogeographic maps, placing Gondwana in the tropics and Avalonia in high southern latitudes, in accordance with the reviewer's comments and the suggested literature.

A final “problem” is interpretation of Cambrian climates as warm even to the poles. At a time of no glaciation, existing studies on oxygen isotopes cannot provide a reasonable paleotemperature. Anyway, the paleogeog map used in the ms. has Baltica in relatively low latitudes during a very warm interval and somehow lacks significant carbonates? And has the same siliciclastic lithofacies as Avalonia shown far to the south?

Both Scotese (2014) and Cuttings (2021) models, respectively based on lithological deposits that are indicative of specific climatic zones (e.g., evaporites, coals, calcretes, or tillites) and climatic simulator in which the CO₂ concentration and albedo are the main variable, are independent to oxygen isotope. However, the position of Gondwana is not revised according to the literature presented in the previous point. By relocating Gondwana in the tropics, the poles should be indeed colder than what the models predicts but the lack of glacial deposits would likely suggest that polar regions were still warm enough to prevent the development of ice caps. In Cuttings (2021)'s model, the average sea surface temperature is estimated to be of 4 to 8°C and could potentially be lowered to 0-4°C if Gondwana is relocated in the tropics.

The text has been adjusted by replacing “warm temperate conditions” by “cool to temperate conditions”.

The positions of Baltica, Avalonia, and Gondwana were adjusted, resolving the facies and lithological succession issues identified by the reviewer.

The paper is frankly difficult to read for people not involved in astrochronology studies. An “Analytical Procedures” section that briefly (an added page) preceeding the “Spectral analysis” section (line 430s) maybe should explain terminology like “inclination metronome,” “time domain analysis,” “obliquity modulation,” etc.

Sorry for the confusion.

We added more precision regarding depth domain, time domain, and metronomes designations in the text.

We also integrated a paragraph in the introduction to briefly introduce cyclostratigraphy to a broader audience.

Here is the added paragraph:

Paleoclimatologic research establishes that quasi-periodic oscillations in the Sun-Earth position, known as Milankovitch cycles, have induced significant variations in Earth’s past climate at different time scales^{e.g., 11,12}. Cyclostratigraphy is a well-established and powerful chronometer, which uses the expression of Milankovitch cycles preserved in the stratigraphic record to refine the geological time scale^{e.g., 12-16}. The principal components of Milankovitch cycles includes short and long orbital eccentricity (~95-135 and ~405 ka), as well as their modulation (~2.4 and ~9 Myr); obliquity (~41 ka), as well as its modulation (~173 ka and ~1.2 Myr); and the precession index (~19 and 23 ka, modulated by orbital eccentricity)^{e.g., 17-20}.

The main steps of the time series protocol for this investigation were added to the material and method section of the manuscript. The detailed protocol targeting a specialized audience remains in the supplementary text.

Reviewer #3

I'm sorry for the delay in providing my review comments. I accepted to review this manuscript because, from the abstract, I understood that it dealt with C chemostratigraphy. However, upon closer reading, I found that the manuscript covers many aspects, most of which are outside my domain of expertise. Therefore, I feel that I am not eligible to provide a robust and comprehensive review.

However, I see that the manuscript adopts a robust multidisciplinary approach to detect the DICE in a new locality, which provides important insights into the Cambrian period, including the possible non-synchronicity of the DICE and the occurrence of biostratigraphic index fossils such as *P. atavus*. From my domain of expertise, I find that the use of $\delta^{13}\text{C}_{\text{org}}$ to detect the DICE is not the most appropriate methodology, since $\delta^{13}\text{C}_{\text{org}}$ may not be synchronous with $\delta^{13}\text{C}_{\text{carb}}$. This is because $\delta^{13}\text{C}_{\text{org}}$ can be altered by the addition of detrital organic carbon, particularly given that the negative excursion identified by the authors occurs directly above an erosional unconformity, which may reflect detrital input.

We agree that relying solely on C_{org} is not the most reliable way to identify a carbon isotope excursion. However, as explained in Supplementary Text 1, the position and shape of the excursion align well with observations from other locations around the world, including those with C_{carb} analyses. Additionally, Álvaro et al. (2010) performed C_{carb} analyses on the Exsulans and Forsemölla limestones that seemingly indicate an excursion between the two. Nevertheless, we would not feel confident basing the correlation of our section on this assumption alone.

Another important point is that land coverage was negligible to nonexistent in the Cambrian period, and Scania was more than 1,000 km away from the continent. Therefore, it is unlikely that the C_{org} was affected by detrital organic carbon. We estimate that the excursion lasted about 750 kyr. If the C_{org} had been affected by detrital organic carbon, we suspect the excursion would have been shorter because ocean mixing occurs in approximately 1 kyr.

You are correct about the synchronicity of the excursion. C_{carb} and C_{org} can sometimes exhibit an offset of up to 500 kyr (e.g., Kump & Arthur, 1999). Consequently, it is possible that the "real" excursion is slightly younger and located on the *P. gibbus*–*P. atavus* boundary. However, if the C_{carb} excursion observed by Álvaro et al. (2010) indeed corresponds to the DICE, then no significant offset would be identified, aligning with the results of studies conducted in South China (Li, D. et al., 2020) and West Gondwana (Jamart et al., 2025).

The paper by Bowyer et al. (2023) on the $\text{C}_{\text{carb-shale}}$ method is an important contribution. The geological setting of our study does not allow us to perform this type of measure as confidently as Bowyer et al. (2023). In Namibia, the presence of several limestone intervals coupled with numerous dated ash beds enabled Bowyer et al. (2023) to verify the accuracy of the $\text{C}_{\text{carb-shale}}$ data. Although Bowyer et al. (2023) conclude that the value of $\delta^{13}\text{C}_{\text{carb-sh}}$ for filling gaps in $\delta^{13}\text{C}_{\text{carb}}$ records remains uncertain, even in shale with high CaCO_3/TOC deposited in mixed carbonate–clastic settings, this issue is even more problematic in Baltica because no $\delta^{13}\text{C}_{\text{carb}}$ reference record is available. Moreover, diagenetic limestone concretions are common in Baltica, and diagenesis could strongly influence the shale carbonate signal, further reducing confidence in $\delta^{13}\text{C}_{\text{carb-sh}}$ data. Even though we do not believe that performing a $\text{C}_{\text{carb-shale}}$ analysis would be suitable for the Alum Shale Formation, we do not reject the idea that the isotopic signal could be adjusted in the future if reliable C_{carb} data becomes available.

(see paper " Bowyer, F., Yilales, M., Wood, R., & Poulton, S. W. (2023). Insights into the terminal Ediacaran marine carbonate record from shale-hosted carbonate carbon isotopes. American Journal of Science, 323.").

Thank you for the article! I was not aware of it.

I believe that if C_{carb} is also measured in the section, it would provide more robust data for correlation with regional sections

We agree that C_{carb} would be better as it is more commonly used to synchronize record worldwide but no carbonates are available in the studied interval.

Line 226: The authors refer to Figure 2 when describing the C_{org} variation with stratigraphy ('from 93.611 to 91.229 m position...'). However, this interval is not illustrated in Figure 2. I recommend homogenizing the metric scale in Figure 2, as the interval '93.611 to 91.229 m' cannot be located. This observation applies to the whole "Carbon stable isotopes ($\delta^{13}\text{C}_{\text{org}}$)"sections."

All my apologies; the text has been corrected as follows: "From 232.616 to 229.975 m (93.611–91.299 m AD)." We also added that the AD depths correspond to the adjusted depths. Thus, readers can easily identify the depths and their corresponding ages on the astronomical timescale in the supplementary materials.

References cited in this document

Álvaro et al. (2010) – Skeletal carbonate productivity and phosphogenesis at the lower–middle Cambrian transition of Scania, southern Sweden – **Geological Magazine**

Arts et al. (2024) – Age and orbital forcing in the upper Silurian Cellon section (Carnic Alps, Austria) uncovered using the WaverideR R package – **Frontiers**

Farrell et al. (2025) – Revising the late Cambrian time scale and the duration of the SPICE event using a novel Bayesian age modeling approach – **GSA Bulletin**

Jamart et al (2025) – The Cambrian ROECE and DICE carbon isotope excursions in western Gondwana (Montagne Noire, southern France): Implications for regional and global correlations of the Miaolingian Series – **Palaeogeography, Palaeoclimatology, Palaeoecology**

Kump & Arthur, 1999 – Interpreting carbon-isotope excursions: carbonates and organic matter – **Chemical Geology**

Laskar et al. (2004) – A long-term numerical solution for the insolation quantities of the Earth – **Astronomy & Astrophysics**

Laskar (2020) – Astrochronology – **GTS 2020**

Lauridsen (2000) – The Cambrian-Tremadoc interval of the Albjåra-1 drill-core, Scania, Sweden – **Master's thesis (unpublished)**

Li, D. et al. (2020) – A paired carbonate–organic $\delta^{13}\text{C}$ approach to understanding the Cambrian Drumian carbon isotope excursion (DICE) – **Precambrian Research**

Li, M et al. (2018) – Sedimentary noise and sea levels linked to land–ocean water exchange and obliquity forcing – **Nature Communications**

Nielsen & Schovsbo (2011) – The Lower Cambrian of Scandinavia: Depositional environment, sequence stratigraphy and palaeogeography – **Earth-Science Reviews**

Nielsen & Schovsbo (2015) – The regressive Early-Mid Cambrian ‘Hawke Bay Event’ in Baltoscandia: Epeirogenic uplift in concert with eustasy – **Earth-Science Reviews**

Sørensen et al. (2020) – Astronomically forced climate change in the late Cambrian – **Earth and Planetary Science Letters**

Wu et al. (2024) – A 650-Myr history of Earth's axial precession frequency and the evolution of the Earth-Moon system derived from cyclostratigraphy – **Science Advances**

Zhao et al. (2022) – Synchronizing rock clocks in the late Cambrian – **Nature Communications**